# THE DUAL POWER OF INTERPRETABLE TOKEN EMBEDDINGS: JAILBREAKING ATTACKS AND DEFENSES FOR DIFFUSION MODEL UNLEARNING

## ABSTRACT

Diffusion models excel at generating high-quality images but can memorize and reproduce harmful concepts when prompted. Although fine-tuning methods have been proposed to unlearn a target concept, they struggle to fully erase the concept while maintaining generation quality on other concepts, leaving models vulnerable to jailbreak attacks. Existing jailbreak methods demonstrate this vulnerability but offer limited insight into how unlearned models retain harmful concepts, limiting progress on effective defenses. In this work, we take one step forward by exploring a linearly interpretable structure. We introduce *SubAttack*, a novel jailbreaking attack that learns an orthogonal set of attack token embeddings, each being a linear combination of human-interpretable textual elements, revealing that unlearned models still retain the target concept through related textual components. Furthermore, our attack is also more powerful and transferable across text prompts, initial noises, and unlearned models than prior attacks. Leveraging these insights, we further propose *SubDefense*, a lightweight plug-and-play defense mechanism that suppresses the residual concept in unlearned models. SubDefense provides stronger robustness than existing defenses while better preserving safe generation quality. Extensive experiments across multiple unlearning methods, concepts, and attack types demonstrate that our approach advances both understanding and mitigation of vulnerabilities in diffusion unlearning.

## 1 INTRODUCTION

Diffusion models (DMs) have recently emerged as a powerful class of generative models, capable of producing diverse and high-quality content such as images (Ho et al., 2020), videos (Khachatryan et al., 2023), and protein structures (Watson et al., 2023). Notably, Text-to-Image (T2I) diffusion models (Rombach et al., 2022; Ramesh et al., 2022a; Saharia et al., 2022; Zhang et al., 2024e;b) have gained significant popularity for their ability to generate high-fidelity images from user-provided text prompts. However, the remarkable generative capabilities of these models also raise significant concerns regarding their safe deployment. For example, users can exploit carefully crafted text prompts to induce these models by generating unethical or harmful content, such as nude or violent images, or copyrighted material (Schramowski et al., 2023).

To address such safety concerns without refiltering the huge dataset and retraining the full model, *Machine Unlearning* (MU) methods have recently been developed for "erasing" a harmful concept directly from the pretrained models. For instance, a wide range of methods (Gandikota et al., 2023; 2024; Zhang et al., 2023; Lyu et al., 2024) seek to unlearn harmful content in pretrained DMs by fine-tuning the model weights (Nguyen et al., 2024). Yet, the key challenge of preserving the generation quality of safe content limits unlearned DMs from removing even a *single concept* completely. This limitation becomes evident under *jailbreaking attacks* (Zhang et al., 2024d; Pham et al., 2024; Chin et al., 2024b; Tsai et al., 2024; Zhuang et al., 2023), which have enforced unlearned DMs to regenerate harmful content. For instance, UnlearnDiff (Zhang et al., 2024d) crafts adversarial discrete text prompts, and CCE (Pham et al., 2024) leverages textual inversion (Gal et al., 2023) to execute jailbreaking attacks in embedding space. Amid the rising popularity of open-source models, and given the risks of insider threats and model leakage, many studies adopt a white-box setting (i.e., full access to model weights) for safety evaluation. These works reveal that unlearned DMs remain

vulnerable, highlighting the urgent need to further *defend* these unlearned DMs by strengthening their robustness against attacks.

It is not surprising that optimization-based, non-interpretable, and worst-case prompt perturbations can jailbreak unlearned DMs. However, despite leveraging white-box access, such approaches provide limited *interpretability*, i.e., a human-understandable explanation of how a model's internal state drives the prediction of its behavior under intervention. Therefore, they offer little insight into how harmful concepts persist within the model, and these attacks fail to offer potential insights for defense strategies. Furthermore, the defense of unlearned DMs remains largely underexplored. For instance, the RECE defense framework (Gong et al., 2024) focuses on improving a specific unlearned model (UCE (Gandikota et al., 2024)) against particularly adversarial attacks (i.e., UnlearnDiff). Extending defenses to a broader range of unlearned models and attack types remains a challenging problem. These gaps motivate our central **question**: *Can we design more human-interpretable jailbreaking attacks that also provide actionable insights for building defenses in unlearned DMs?*

Our work tackles this fundamental question by exploring underlying linear structures, taking advantage of the white-box setting. We introduce an effective, human-interpretable *subspace attack method (SubAttack)*, which further inspires a *subspace defense strategy (SubDefense)* broadly applicable to various unlearned models and attacks. The core idea is to learn an orthogonal set of attack token embeddings within the unlearned model for the harmful concept. Inspired by prior works (Chefer et al., 2024; Park et al., 2023a; Cunningham et al., 2023), we optimize each attack embedding as a nonnegative linear combination of embeddings of existing concepts, and interpret the concept through the linear decomposition. Leveraging our approach, we show that unlearned DMs associate the harmful concept with mixtures of other hidden concepts, thus retaining unintended harmful regeneration capabilities. These insights motivate our defense mechanism, which further mitigates the harmful concept from unlearned DMs by removing the learned attack token embeddings through orthogonal subspace projection.

Compared to prior methods, our SubAttack demonstrates strong empirical performance of efficiency and effectiveness, showing stronger transferability across text prompts, initial noises, and unlearned models. Our defense strategy can be seamlessly integrated into various unlearned models, improving robustness against different jailbreaking attacks while preserving higher generation quality than the baseline defense method (Gong et al., 2024). A comprehensive discussion of related works is in **App. A**. In summary, this work makes the following **contributions**:

- **Interpretable attack via linear structure.** We propose *SubAttack*, which learns an orthogonal set of token embeddings under a linear structure. These embeddings can be interpreted in a bag-of-words fashion, revealing how the residual concept is still retained in unlearned DMs.
- **Effective and transferable attack.** *SubAttack* achieves higher ASR than existing baselines across diverse concepts and unlearned models, while also transferring reliably across prompts, initial noise, and models, exposing critical vulnerabilities in current unlearning methods.
- **Subspace defense inspired by the linear interpretable structure.** Leveraging this linear structure, we propose *SubDefense*, which projects out attack token directions to eliminate residual concepts. Our SubDefense offers versatile, reliable protection while preserving generation quality.

## 2 PRELIMINARIES AND PROBLEM STATEMENT

### 2.1 PRELIMINARIES

**Overview of Latent Diffusion Models (LDMs).** T2I diffusion models have recently gained popularity for their ability to generate desired images from user-provided text prompts. Among these various T2I models, LDMs (Rombach et al., 2022) is the most widely deployed DM, and has therefore become the primary focus of current machine unlearning methods. As shown in **Fig. 1**, for a given text prompt $p$, LDM first encodes $p$ using a pretrained CLIP text encoder (Radford et al., 2021) $f(\cdot)$ to obtain the text embedding $c = f(p)$. Then, the generation process begins by sampling a random noise $z_T \sim \mathcal{N}(0, 1)$ in the latent space. After that, LDM progressively denoises $z_T$ conditioned on the context $c$ until the final clean latent $z_0$ is achieved. Specifically, for each timestep $t = T, T-1, \ldots, 1$, its denoising UNet, $\epsilon_\theta(z_t \mid c)$, predicts and removes the noise to obtain a cleaner latent representation $z_{t-1}$. The clean latent $z_0$ is then decoded to an image with a pretrained

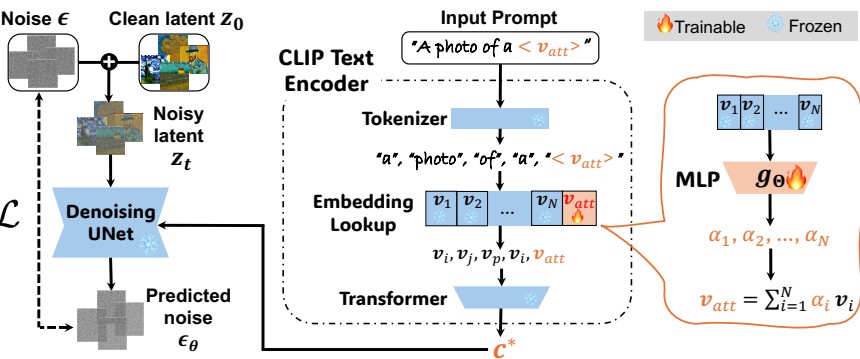

Figure 1: **Learning one interpretable attack token embedding.** The learning process of one attack token embedding $v_{att}$ for the concept "Van Gogh" is visualized. Blue parts represent the frozen unlearned LDM, where, for simplicity, we omit the image encoder and decoder. In orange parts, it illustrates the learning mechanism for optimizing an MLP network to produce $v_{att}$, which is a linear combination of the existing token embeddings.

image decoder. To train the denoising UNet $\epsilon_{\theta}(z_t \mid c)$ in LDM, the denoising error is minimized:

$$\mathcal{L} = \mathbb{E}_{(z,c),t,\epsilon \sim \mathcal{N}(0,1)} \left[ \| \epsilon - \epsilon_{\theta} \left( z_t \mid c \right) \|_2^2 \right],$$ (1)

where $z$ is the clean image latent encoded by a pretrained image encoder and $c$ is the corresponding text embedding. Here, $z_t = \sqrt{\alpha_t} z + \sqrt{1 - \alpha_t} \epsilon$ is the noisy image latent at timestep $t$, and $\alpha_t > 0$ is a predefined constant.

**CLIP text encoder and the token embedding space.** To control the generation process, a key component of LDMs is the pretrained CLIP text encoder $f(\cdot)$. As illustrated in **Fig. 1**, the CLIP text encoder consists of three main components:

- **Tokenizer:** This module splits the text prompt $p$ into a sequence of tokens, which can be words, sub-words, or punctuation marks. Each token is assigned a unique token ID from the CLIP text encoder's predefined vocabulary.
- **Token Embeddings:** These token IDs (e.g., $[i, j, \cdots]$) are then mapped to corresponding token embeddings $v_i \in \mathbb{R}^d$ stored in the token embedding table. This process generates a sequence of token embeddings $[v_i, v_j, \cdots]$.
- **Transformer Network:** This network processes the sequence of token embeddings and encodes them into the final text embedding $c$ that can guide the image generation process in LDMs.

Through optimizing Eq. (1), LDM learns to associate activations in the text encoder with concepts in the generated images. Prior research has explored controlling generated content through manipulating activations in the text encoder. In particular, it has been identified that the token embedding space $v$ plays a vital role in content personalization, where a single text embedding can represent a specific attribute (Gal et al., 2023) and the token embedding space can be utilized for linear decomposition of concepts (Chefer et al., 2024). Leveraging the expressiveness and interpretability of the token embedding space, we propose both jailbreaking attack and defense mechanisms, and discuss the problem setup in the following.

## 2.2 PROBLEM STATEMENT AND SETUP

***Jailbreaking attacks*** are designed to evaluate the robustness of unlearned LDMs. Most existing diffusion unlearning studies focus on removing *a single target concept* from each model. For example, given a prompt $p =$ "a photo of a [target concept] ...", an unlearned LDM for this concept is expected to have difficulty generating the corresponding images. A jailbreaking attack, given an unlearned LDM as the *victim model*, aims to manipulate the prompt to make the model regenerate the unwanted concept. There are majorly two kinds of attack setups: (*i*) Adversarial prompt-based attacks (Zhang et al., 2024d; Chin et al., 2024b; Tsai et al., 2024; Zhuang et al., 2023) optimize an *adversarial text prompt* $p_{att}$ and append it to $p$. (*ii*) Embedding-based attacks (Pham et al., 2024) learn an *attack token embedding* $v_{att}$, register it as a new token $< v_{att} >$, and modify the prompt by replacing the [target concept] with this token. Our attack follows the second setup, but is explicitly designed to

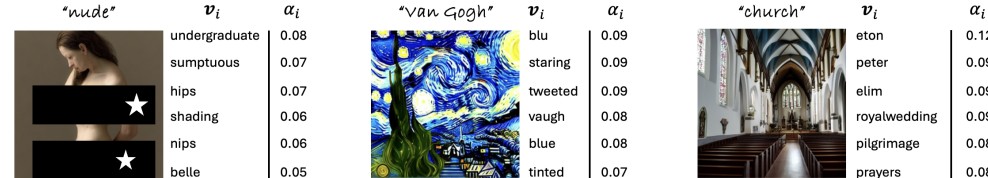

Figure 2: **Interpreting the attack token embeddings for concept "nudity", "Van Gogh", and "church".** Tokens with the largest $\alpha_i$ are words associated with the target concept. For example, top tokens for "church" are activities conducted in the church, or names from the Bible.

be interpretable through linear constraints while achieving stronger attack performance. Moreover, apart from access to the unlearned LDM, existing attacks generally require either the original LDM or images containing the target concept; in our setup, we assume access to the images ($z_0$ in **Fig. 1**).

*Defense*, in contrast, seeks to protect an *unlearned LDM* from new jailbreaking attacks. Once a defense strategy is applied, it should prevent the model from regenerating harmful concepts even under *future attacks*, while preserving its ability to generate harmless content. For example, RECE (Gong et al., 2024) further modifies the denoising UNet of the unlearned model UCE (Gandikota et al., 2024) to defend against adversarial attacks (Zhang et al., 2024d). In this work, we propose a defense strategy that safeguards the token embedding space and can be seamlessly integrated into existing unlearned LDMs. Our objective is to provide a broadly applicable defense mechanism that enhances robustness across diverse unlearned models when confronted with new attacks.

*Interpretability* refers to providing a compact, human-understandable description of how a model's internal components drive its behavior (Chefer et al., 2024; Zou et al., 2023; Bereska & Gavves, 2024). Such a description is crucial as it allows testable predictions under controlled interventions. In our setting, interpretability means that attack embeddings can be explained as recognizable words or semantic units rather than opaque vectors. While many existing methods are empirically effective, they lack such interpretability, making it difficult to understand how harmful concepts persist or how to control the robustness of unlearned models. Our work addresses this gap by developing attack and defense methods grounded in a linear, interpretable structure.

**Notations.** Before introducing our method, we define the following projection operators. Specifically, given vector $z$, for a vector $v$, let $\mathrm{Proj}_{v}(z)$ denote the projection of $z$ onto $v$. For a matrix $V$, let $\mathrm{Proj}_{V}(z)$ denote the projection of $z$ onto the subspace spanned by the columns of $V$. Formally, these operators are given by

$$\mathrm{Proj}_{v}(z) := \frac{vv^{\top}}{\|v\|_2^2}z, \ \mathrm{Proj}_{V}(z) := V(V^{\top}V)^{-1}V^{\top}z.$$

## 3 Subspace Attacking and Defending Methods

This section introduces our subspace attacking and defending methods for LDMs. In Sec. 3.1, we explore the token embedding space to develop an interpretable and effective attack method (SubAttack) by learning a sequence of attack token embeddings orthogonal to each other. SubAttack further inspires us to propose a defense strategy (SubDefense) in Sec. 3.2, by orthogonal subspace projection of learned attack token embeddings, which can effectively defend against various jailbreaking attacks.

### 3.1 Subspace Attacking: *SubAttack*

Before we introduce our subspace attacking method, let us build some intuition of how to learn a single interpretable attack token embedding $v_{\mathtt{att}} \in \mathbb{R}^d$. Based on this, we will then show how to iteratively learn a sequence of orthogonal attack token embeddings through *deflation*, i.e., removing already computed embeddings.

### 3.1.1 Single-Token Embedding Attack

We aim to learn a token embedding $v_{\texttt{att}} \in \mathbb{R}^d$ as a non-negative linear combination of existing token embeddings $v_i$ in the CLIP vocabulary $\mathcal{V}$ as follows:

$$v_{\texttt{att}} = \sum_{i=1}^{N} \alpha_i v_i, \quad \alpha_i = g_\Theta(v_i) \geq 0, \tag{2}$$

where $N$ is the total size of the original CLIP vocabulary, and $v_i$, $i = 1, 2, \ldots, N$, are original CLIP token embeddings within $\mathcal{V}$. Non-negative $\alpha_i$ are parameterized via a multi-layer perceptron (MLP) network $g_\Theta(\cdot) : \mathbb{R}^d \mapsto \mathbb{R}^+$ with ReLU activation. This is inspired by recent work (Chefer et al., 2024) on language models. To learn $v_{\texttt{att}}$, we **optimize the loss** $\mathcal{L}$ in Eq. (1) with respect to the **parameter $\Theta$ of the MLP**, while freezing all the other components. As illustrated in **Fig. 1**, during training we enforce the training data pairs $(z, c^*) \sim \mathcal{D}$ to satisfy the following constraints: (*i*) $z$ is the latent image containing the target harmful concept. (*ii*) $c^*$ is the text embedding for the text prompt $p$, and $p$ contains the new special token $< v_{\texttt{att}} >$ whose token embedding is $v_{\texttt{att}}$.

**Remarks.** The non-negative constraint in Eq. (2) is inspired by prior works on linear representation hypothesis and linear feature decomposition (Chefer et al., 2024; Zhou et al., 2018; Cunningham et al., 2023; Park et al., 2023a) that "negative concepts are not as interpretable as positive concepts." In this way, the target concept can be viewed as a combination of top-weighted (i.e., having largest $\alpha_i$) concepts in $\mathcal{V}$. **Fig. 2** illustrates the identified sets of human-interpretable concepts for different target concepts (e.g., nudity, Van Gogh, church) in unlearned LDMs. Additionally, we provide analysis on the sparsity of $\alpha_i$ in **App. F**. Now, we introduce how a set of attack token embeddings are learned.

### 3.1.2 Subspace Token Embedding Attacks

Compared with learning a single attack token embedding $v_{\texttt{att}}$, it is more powerful to learn a set of diverse attacks $\{v_{\texttt{att},k}\}_{k=1}^{K}$ ($m \leq d$) that can attack the same target concept, as outlined in **Algorithm 1**. We enforce orthogonality on $\{v_{\texttt{att},k}\}_{k=1}^{K}$ to promote diversity and improve attack effectiveness (see ablations in **App. E.1**).

Such a set of orthogonal token embeddings $\{v_{\texttt{att},k}\}_{k=1}^{K}$ is learned through deflation, sharing similar spirits with classical numerical methods such as orthogonal matching pursuit (Tropp & Gilbert, 2007). Specifically, suppose the first attack token embedding $v_{\texttt{att},1}$ is identified following Sec. 3.1.1 by optimizing an MLP $g_{\Theta_1}$,

---

**Algorithm 1** Learning Attack Token Embeddings

1: **Input:** victim model with CLIP token embeddings $[v_{1,1}, \ldots, v_{N,1}]$, total iterations $K$
2: **Output:** $[v_{\texttt{att},1}, \ldots, v_{\texttt{att},K}]$
3: **for** $k = 1, \ldots, K$ **do**
4: $\quad$ Optimize the MLP $g_{\Theta_k}$
5: $\quad \alpha_{i,k} \leftarrow g_{\Theta_k}(v_{i,k})$
6: $\quad v_{\texttt{att},k} \leftarrow \sum_{i=1}^{N} \alpha_{i,k} v_{i,k}$
7: $\quad$ **for** $i = 1, \ldots, N-1$ **do**
8: $\quad\quad v_{i,k+1} \leftarrow v_{i,k} - \text{Proj}_{v_{\texttt{att},k}}(v_{i,k})$
9: $\quad$ **end for**
10: **end for**

---

we then "eliminate" the target concept $v_{\texttt{att},1}$ from the whole vocabulary $\mathcal{V}$ via orthogonal projection:

$$v_{i,2} = v_{i,1} - \text{Proj}_{v_{\texttt{att},1}}(v_{i,1}), \quad \forall i \in [N]. \tag{3}$$

Here, $v_{i,1} \equiv v_i \in \mathcal{V}$ are the original embeddings for all $i \in [N]$. Eq. (3) makes sure *all* the updated $v_{2,i}, \ldots, v_{2,N}$ are orthogonal to $v_{\texttt{att},1}$. With the new $\mathcal{V}_2 = \{v_{2,i}\}_{i=1}^{N}$, we can learn a second attack token embedding $v_{\texttt{att},2} = \sum_{i=1}^{N} \alpha_{i,2} v_{i,2}$, $\alpha_{i,2} = g_{\Theta_2}(v_{i,2}) \geq 0$, then $v_{\texttt{att},2}$ **is ensured to be orthogonal to $v_{\texttt{att},1}$**. Here, $g_{\Theta_2}$ is another MLP optimized in the same way as $g_{\Theta_1}$. As such, we can repeat the procedure for $K$ times to learn and construct a set of orthogonal attack token embeddings $\{v_{\texttt{att},k}\}_{k=1}^{K}$, and use each of them to attack the same target concept. In practice, during attacking, we choose $K = 5$, which delivers strong attack performance while keeping the method efficient (see ablation studies in **App. E.1**).

## 3.2 Subspace Defending: *SubDefense*

Our SubAttack reveals that combinations of related hidden concepts can represent the target concept in an unlearned LDM through a linear composition. This insight motivates us to design a defense strategy within the same linear framework. Our intuition is to remove these identified concept representations from unlearned models through orthogonal projection, thereby making them more

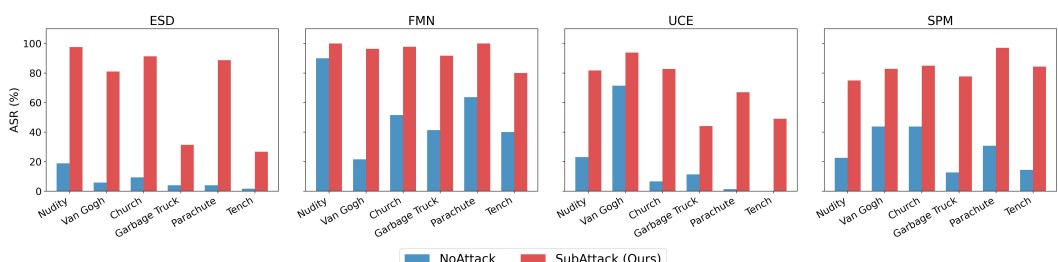

Figure 3: **SubAttack jailbreaks various concepts (NSFW, style, objects) across different unlearned models (ESD, FMN, UCE, SPM).** It consistently reveals the residual vulnerabilities in these models.

robust to various jailbreaking attacks. Concretely, because linearly composed concepts become more difficult to recover, this is achieved by projecting onto the null space of the learned subspace attacks.

More specifically, suppose we have learned a set of attack token embeddings $\{v_{\text{att},k}\}_{k=1}^K$ for a target concept through SubAttack outlined in Sec. 3.1, then let us rewrite

$$V_{\text{att}} = [v_{\text{att},1} \quad v_{\text{att},2} \quad \cdots \quad v_{\text{att},K}] \in \mathbb{R}^{d \times K}.$$

This $V_{\text{att}}$ is learned in an unlearned diffusion model whose CLIP token embedding vocabulary is $\mathcal{V} = \{v_i\}_{i=1}^N$. The proposed defense will "block" the subspace spanned by $V_{\text{att}}$ through orthogonal projection. Each token embedding $v_i$ in $\mathcal{V}$ will be updated as follows:

$$v_{\text{def},i} = v_i - \text{Proj}_{V_{\text{att}}}(v_i), \quad \forall i \in [N]. \tag{4}$$

For *UnlearnDiff* (Zhang et al., 2024d) and *SubAttack*, their learned jailbreaking attack prompts or embeddings are based on the unlearned LDM's vocabulary. Hence, we will update the unlearned LDM by applying Eq. (4) to complete the defense. After that, *new* UnlearnDiff and SubAttack attacks can take place on the updated model, but have lower ASR (Sec. 5). For *CCE* (Pham et al., 2024), which learns an attack token embedding $v_{\text{att}}$ with no constraints related to the unlearned LDM's vocabulary $\mathcal{V}$, simply applying Eq. (4) is not enough. Hence, additionally, for *new* $v_{\text{att}}$ learned by CCE, $v_{\text{def}} = v_{\text{att}} - \text{Proj}_{V_{\text{att}}}(v_{\text{att}})$ is applied. In SubDefense, we name $K$ as the number of blocked tokens.

## 4 EXPERIMENTS ON SUBATTACK

This section first provides a deeper analysis of the interpretable tokens it identifies, and leverages this interpretability to reveal how current unlearned LDMs still conceal target concepts. We then demonstrate through extensive experiments that SubAttack is not only more effective than baseline attacks but also highly transferable.

### 4.1 SETTINGS

(*i*) **Victim Models.** We evaluate SubAttack on a broad set of diffusion-model unlearning methods commonly used in prior jailbreak studies, including ESD (Gandikota et al., 2023), FMN (Zhang et al., 2023), and UCE (Gandikota et al., 2024), as well as more recent or complementary settings such as SPM (Lyu et al., 2024), MACE (Lu et al., 2024), SA (Heng & Soh, 2023), AC (Kumari et al., 2023), SalUn (Fan et al., 2023), and EraseDiff (Wu et al., 2024). Following prior work (Zhang et al., 2024d), all unlearned models are fine-tuned from Stable Diffusion v1.4 (Rombach et al., 2022).

(*ii*) **Concepts and Dataset.** We perform jailbreaking attacks on representative concept categories in prior diffusion unlearning: "nudity" for NSFW, "Van Gogh" for style, and objects such as "church", "garbage truck", "parachute", "tench", "airplane", etc. Following UnlearnDiff (Zhang et al., 2024d), we construct 300–900 (prompt, seed) pairs per concept, with at least 10 seeds per prompt to reduce randomness and evaluate transferability. Our dataset is $\approx 6\times$ larger than UnlearnDiff's, enabling a more reliable assessment.

(*iii*) **Attack and Evaluation.** For each concept, SubAttack learns $K = 5$ token embeddings $\mathbf{v}_{\text{att},k}$, and an attack is successful if any embedding regenerates the target concept. Further ablations on $K$,

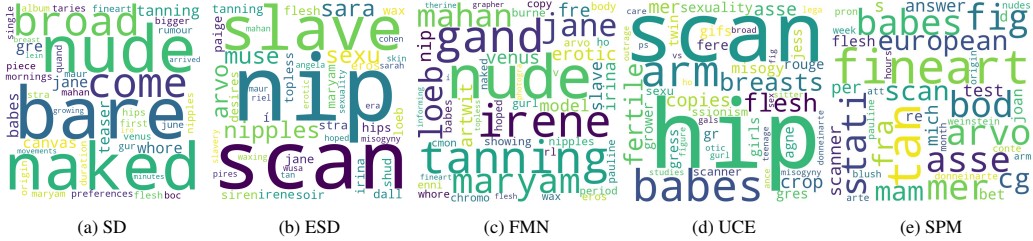

|  (a) SD | (b) ESD | (c) FMN | (d) UCE | (e) SPM |

Figure 5: **Interpreting the subspace of attack token embeddings for concept "nudity" across different models.** (a) The original LDM (i.e., SD) majorly relates it to **explicit** synonyms. (b-e) Unlearned LDMs more heavily associate it with **implicit** concepts.

orthogonality, and size of vocabulary are in **App. E.1**, with sparsity analysis in **App. F**. We report attack success rate (ASR) using pretrained classifiers following (Zhang et al., 2024d): NudeNet (Platelminto, 2024) for NSFW, a WikiArt-finetuned model for style, and an ImageNet-pretrained ResNet-50 for objects.

(*iv*) **Baselines.** We compare SubAttack against three baselines: NoAttack (original prompts without jailbreak), UnlearnDiff (Zhang et al., 2024d), and CCE (Kumari et al., 2023). UnlearnDiff and CCE are reproduced with their original settings but unified under our dataset (e.g., UnlearnDiff optimizes an adversarial prompt per (prompt, seed) pair). We provide more experiment details in **App. B.1**.

## 4.2 INTERPRETABILITY OF PROPOSED SUBATTACK METHODS

We analyze the embeddings $\{\boldsymbol{v}_{\mathrm{att},k}\}_{k=1}^{K}$ to examine how target concepts persist in unlearned LDMs. For each $\boldsymbol{v}_{\mathrm{att},k}$, we extract the top-50 highest-weighted tokens, stem and lemmatize them, and visualize the most frequent ones with WordCloud. The same procedure is applied to the original SD for comparison. We present key examples and findings below, with more results in **App. C.1**.

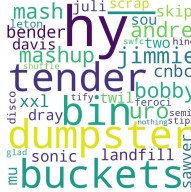

(*i*) **SubAttack enables learned embeddings understandable to humans.** The resulting tokens reveal meaningful and positively associated concepts rather than random noise. We observe sexualized terms for the NSFW concept (e.g., "slave", "babes") in **Fig. 5**, cross-lingual variants for church (e.g., "kirk" in Scottish English) in **Fig. 10**, and key painting elements for Van Gogh (e.g., "oats", "night") in **Fig. 11**. These findings verify that the learned embeddings are directly interpretable, providing a clear semantic view of what remains in unlearned models.

Figure 4: **ESD for "garbage truck"**.

(*ii*) **SubAttack shows how stronger unlearning mutes keywords yet leaves hidden clues.** Based on the human-understandable embeddings, we can directly compare original SD and unlearned LDMs to observe a clear progression in how concepts persist. As shown in **Fig. 5**, SD relies on obvious keywords (e.g., "nude," "naked"), weakly unlearned models retain both obvious and hidden terms (e.g., "tanning," women's names), and strongly unlearned models suppress the obvious ones but still retain hidden associations (e.g., "slave," "nip," "babes"). A similar effect appears in other concepts as well in **App. C.1**. Notably, even a strong unlearned "garbage truck" model with only 4% NoAttack ASR still surfaces terms like "dumpster," "bin," and "landfill" (**Fig. 4**). These findings show that unlearning reduces surface-level cues but does not eliminate deeper associations, providing insights unavailable from non-interpretable attacks.

(*iii*) **SubAttack measures how closely the remaining concept matches the original concept.** Beyond visualization, SubAttack provides a quantitative way to assess similarity. Using CLIP similarity between attack tokens and the target concept (**Tab. 1**), we find that weaker unlearning models (e.g., UCE for "Van Gogh," FMN/SPM for "church") retain tokens more semantically aligned with the original concept and also exhibit higher ASR under NoAt-

Table 1: **CLIP similarity** between residual and original explicit concept across unlearned LDMs.

| Concept | ESD | FMN | UCE | SPM |
|---|---|---|---|---|
| Van Gogh | 0.61 | 0.61 | 0.74 | 0.67 |
| Church | 0.76 | 0.85 | 0.79 | 0.82 |

tack (**Fig. 3**). These results suggest that SubAttack can be used to quantify how much of a concept explicitly remains in unlearned models.

Table 3: **Attack performance of various jailbreaking methods**, measured by ASR (%) over 900 prompts for each concept across various unlearned models, average computation time for attacking one image, and other features. Best results are highlighted in **bold**.

| Concepts: | Nudity | | | | Van Gogh | | | | Church | | | | Time per Data (s) ↓ | Interp-retable | Inspire Defense |
|---|---|---|---|---|---|---|---|---|---|---|---|---|---|---|---|
| | ASR (%) ↑ | | | | | | | | | | | | | | |
| Victim Models: | ESD | FMN | UCE | SPM | ESD | FMN | UCE | SPM | ESD | FMN | UCE | SPM | | | |
| NoAttack | 18.78 | 90.00 | 23.00 | 22.56 | 5.78 | 21.56 | 71.44 | 43.78 | 9.33 | 51.56 | 6.55 | 43.78 | NA | NA | NA |
| UnlearnDiff | 51.11 | **100.00** | 78.22 | **83.33** | 40.94 | **100.00** | **100.00** | 53.49 | 51.74 | 35.33 | 61.67 | 53.67 | 906.6 | ✗ | ✗ |
| CCE | 85.11 | 98.33 | 77.22 | 78.33 | 75.22 | 93.33 | 95.67 | 81.67 | 82.00 | **97.78** | 81.89 | 76.67 | **11.4** | ✗ | ✗ |
| SubAttack (Ours) | **97.56** | **100.00** | **81.67** | 74.89 | **81.00** | 96.33 | 98.33 | **82.78** | **91.33** | **97.78** | **82.67** | **84.89** | 54.2 | ✓ | ✓ |

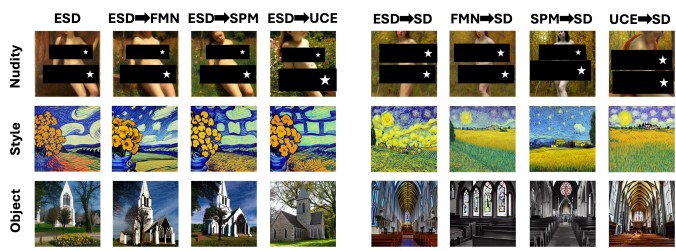

(a) Transfer across unlearned models      (b) Transfer to original stable diffusion

(*iv*) **SubAttack shows the remained concept is inherited from the original SD.** SubAttack embeddings remain effective when transferred back into the original SD. Transfer ASR is consistently above 80% across all concepts and models (See **App. C.1 Tab. 6**; visualized in **Fig. 6 (b)**), suggesting that residual associations in unlearned

Figure 6: **Transfer attack** token embeddings learned by SubAttack to different unlearned models or to the original diffusion model.

models are inherited from SD rather than independently formed. These inherited associations are likely a key reason unlearned models continue to generate harmful content.

### 4.3 EFFECTIVENESS OF PROPOSED SUBATTACK METHODS

(*i*) **SubAttack is an efficient global attack.** UnlearnDiff is a local attack that optimizes an adversarial prompt for each (prompt, seed) pair, which is time-consuming. In contrast, SubAttack learns global attack token embeddings that generalize across prompts and seeds. As shown in **Fig. 3**, SubAttack's global embeddings can jailbreak diverse concepts across hundreds of prompts and seeds. Consequently, SubAttack requires substantially less time per data point on average (see **Tab. 3**).

(*ii*) **SubAttack is highly effective.** As shown in **Tab. 3**, SubAttack exhibits strong attack success rates (ASR). Notably, even as a local attack, UnlearnDiff frequently underperforms SubAttack; for instance, on the "church" concept across multiple unlearned models. While CCE learns unconstrained attack embeddings with commendable performance, it lacks interpretability. In contrast, SubAttack enforces explicit linear structures, which not only enhance performance but also intrinsically enable interpretability. Furthermore, **Fig. 7** illustrates SubAttack's superior fidelity to text prompts. It can faithfully integrate a nude figure into diverse backgrounds such as snowy parks, jungles, and woods, demonstrating precise compositional control. Additional visualizations are provided in **App. H**.

(*iii*) **SubAttack is transferable across different unlearned LDMs.** The attack token embeddings learned by SubAttack transfer robustly between unlearned LDMs. As shown in **Fig. 6 (a)**, embeddings learned via SubAttack on the ESD model are directly transferred to attack FMN, SPM, and UCE. All three concept types, nudity, style, and object, can be successfully transferred to these target models with high ASR. We further compare the transfer ASR of SubAttack against other baselines in **Tab. 2** (more results in **Tab. 14** in **App. C.2**), where we transfer the token embeddings from CCE and the adversarial prompts from UnlearnDiff to other victim models accordingly. SubAttack consistently achieves the highest transfer ASR across different models and concepts. This strong transferability matches the finding that SubAttack identifies embeddings inherited from the original SD model (**Sec. 4.2**).

Table 2: **Transfer attack performance of various jailbreaking methods** from ESD to other models across different concepts, measured by ASR (%).

| Concepts: | Nudity | | | Van Gogh | | | Church | | |
|---|---|---|---|---|---|---|---|---|---|
| Victim Models: | FMN | UCE | SPM | FMN | UCE | SPM | FMN | UCE | SPM |
| NoAttack | 90.00 | 23.00 | 22.56 | 21.56 | 71.44 | 43.78 | 51.56 | 6.55 | 43.78 |
| UnlearnDiff | 93.33 | 41.33 | 38.22 | 12.78 | 64.00 | 47.11 | 6.19 | 13.33 | 58.00 |
| CCE | 93.00 | 18.33 | 37.56 | 72.33 | 43.56 | 81.33 | 91.00 | 70.11 | **92.78** |
| SubAttack (Ours) | **96.89** | **77.00** | **80.44** | 72.67 | **88.89** | **86.89** | **92.89** | **83.77** | 92.00 |

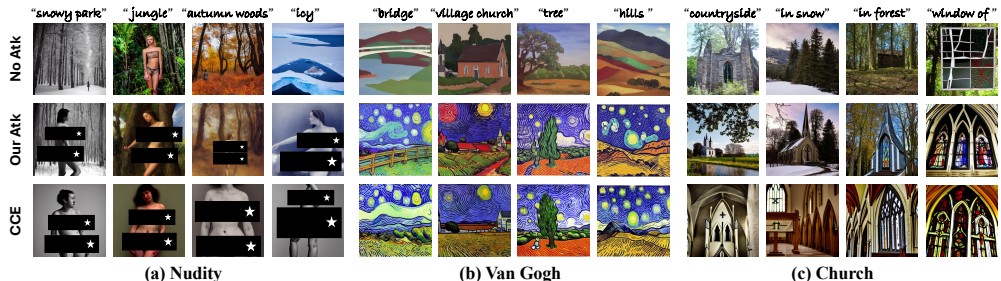

Figure 7: **SubAttack can generate the target concepts with high ASR while aligning with original text prompts**. For example, our attack generates nude women with different backgrounds while CCE fails to generate the correct backgrounds.

## 5 EXPERIMENTS ON SUBDEFENSE

Having established the effectiveness of SubAttack, we next demonstrate the SubDefense method inspired by our attack. We integrate SubDefense into existing unlearned models and assess its robustness. Comprehensive results show that SubDefense offers a more versatile and robust defense than baseline methods, while better preserving generation quality on safe prompts.

### 5.1 SETTINGS

(*i*) **Basics.** SubDefense is plugged into UCE, ESD, FMN, and SPM for concepts "nudity", "Van Gogh", and "church" using our constructed dataset by default. To compare with the baseline RECE framework that defends UCE, we apply SubDefense onto UCE with

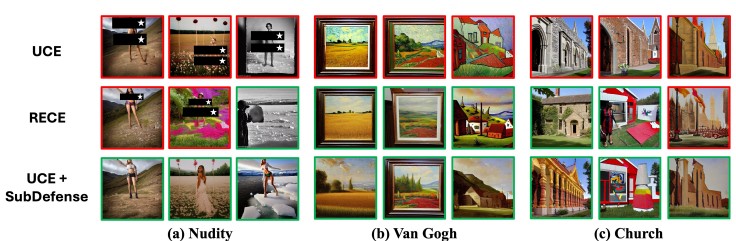

Figure 8: **Defending UCE** using RECE or SubDefense across various concepts.

20 blocked tokens, which already yields better results. In other cases, we use the default setting of 100 blocked tokens. (*ii*) **Metrics.** To assess defense effectiveness, new jailbreaking attacks are conducted after applying defenses, and the corresponding ASR is reported. SubAttack with $K = 5$ is used consistently before and after defense to ensure a fair comparison. Additionally, the generative quality of the defended unlearned models is evaluated on the MSCOCO-10k dataset (Lin et al., 2014; Zhang et al., 2024c) using FID and CLIP scores (Hessel et al., 2021). Further details are in **App. B.2**.

### 5.2 PERFORMANCE OF SUBDEFENSE

(*i*) **SubDefense demonstrates a stronger defense.** We compare SubDefense with RECE (Gong et al., 2024), which is proposed to defend UCE against adversarial attacks. As shown in **Tab. 4**, SubDefense achieves lower ASR, while also attaining lower FID and higher CLIP scores on COCO-10k across three categories of concepts, indicating stronger robustness and better preservation of safe generation quality (**Fig. 8**, **Fig. 9**). More visualizations are provided in **App. G**. In particular, for the 'Van

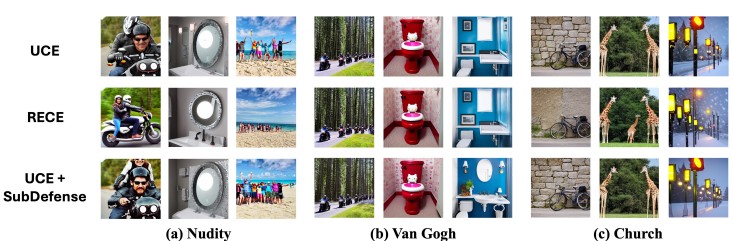

Figure 9: **Safe image generation** after applying RECE or SubDefense.

Gogh' concept, which is closely tied to 'blue' and 'star,' SubDefense preserves these benign elements, demonstrating that it goes beyond naive blocking of all related tokens.

Table 4: **SubDefense is more robust than baseline RECE in defending three concepts on UCE against UnlearnDiff or our SubAttack, while preserving better generative quality.**

| Metrics: | UnlearnDiff ASR ↓ | | SubAttack ASR ↓ | | COCO-10k FID ↓ | | COCO-10k CLIP ↑ | |
|---|---|---|---|---|---|---|---|---|
| Scenarios: | SubDefense | RECE | SubDefense | RECE | SubDefense | RECE | SubDefense | RECE |
| Nudity | **73.55%** | 76.44% | **34.11%** | 62.44% | **17.51** | 17.57 | **30.70** | 30.07 |
| Van Gogh | **52.78%** | 61.67% | **29.44%** | 84.44% | **16.64** | 17.11 | **30.94** | 30.08 |
| Church | **39.78%** | 50.78% | **5.22%** | 80.33% | **17.41** | 17.41 | **30.86** | 30.07 |

(*ii*) **SubDefense is robust across attacks, models, and concepts.** On ESD "nudity," SubDefense lowers ASR against UnlearnDiff, SubAttack, and CCE (**Tab. 5**), showing its ability to defend against diverse jailbreak strategies.

Table 5: **SubDefense can defend ESD against different kinds of attacks.**

| Metrics: | Nudity ASR | | | | CLIP | FID |
|---|---|---|---|---|---|---|
| | NoAttack | UnlearnDiff | CCE | SubAttack | | |
| ESD | 18.11% | 51.11% | 85.11% | 97.56% | 30.13 | 18.23 |
| ESD+SubDefense | 0.0% | 4.56% | 75.67% | 42.33% | 29.58 | 19.20 |

Extended results confirm its effectiveness across other unlearned models (FMN and beyond) and concepts (I2P and beyond) in **Apps. D.2 and D.3**. We also include exploratory results on a classic black-box attack and on the original SD in **App. D.4**.

(*iii*) **SubDefense offers a complementary linear refinement against CCE.** Although a recent nonlinear unlearning method, STEREO Srivatsan et al. (2024), shows improved robustness, CCE remains one of the most challenging white-box attacks to defend for most existing unlearned models. Our goal is not to replace such nonlinear pipelines, but to test—through a plug-and-play refinement strategy inherited from our interpretable diagnosis—whether the *linear* residual structure revealed by SubAttack can further reduce CCE success. Within this linear framework, SubDefense consistently lowers CCE ASR. As shown in **App. E.2**, projecting more directions reduces ASR from 85.11% to 8.89%. This establishes SubDefense as a simple, interpretable refinement, while clarifying the limits of linear defenses and motivating future exploration of potential nonlinear residual structures (**App. I**).

## 6 CONCLUSION

This paper introduces SubAttack, a new jailbreaking method that learns token embeddings capable of regenerating harmful concepts in unlearned diffusion models. Beyond its effectiveness, SubAttack is interpretable: it reveals that unlearned models still retain a broad residual subspace where target concepts are embedded through human-interpretable associations. The attack also shows strong transferability across prompts, noise inputs, and models, exposing deeper vulnerabilities in current unlearning techniques. Building on these insights, we propose SubDefense, a plug-and-play mechanism that disrupts residual subspaces to defend against diverse attacks while preserving generation quality. Together, our findings highlight the urgent need for more robust unlearning methods and provide actionable directions for strengthening the safety of generative diffusion models.

## 7 ETHICS STATEMENT

This work examines the vulnerabilities of diffusion models to jailbreaking attacks, where models regenerate concepts they were intended to unlearn, and introduces corresponding defenses. While the proposed SubAttack could be misused to bypass safeguards and generate harmful content, its purpose here is diagnostic: to expose residual associations in unlearned models and motivate stronger defenses. All experiments were conducted on research models and standard benchmark concepts (e.g., nudity, objects, artistic styles) under controlled conditions, consistent with prior unlearning literature.

We emphasize that our contributions are intended to improve model safety, not to enable harmful applications. By pairing attack analysis with defense strategies, named SubDefense, our work seeks to inform more robust unlearning methods and responsible deployment of generative models. Nonetheless, we recognize that no defense mechanism can guarantee absolute protection, and further safeguards will be necessary in real-world use.

## 8 REPRODUCIBILITY STATEMENT

We have taken multiple steps to ensure reproducibility of our work:

**Code and Implementation.** We will release the full codebase, including data preprocessing, attack and defense implementations, and evaluation scripts, upon publication. Our implementation is based on PyTorch and HuggingFace Diffusers.

**Datasets.** Following prior unlearning works, we construct concept-specific datasets (nudity, objects, artistic styles) using public prompts and seeds. Details are provided in App. B.1. All constructed data will be released.

**Hyperparameters.** Full hyperparameter settings for attack and defense methods (e.g., MLP architecture, learning rates, optimizer, $K$, vocabulary size, number of blocked tokens) are reported in the main text and appendix.

**Evaluation.** We adopt publicly available classifiers (NudeNet, WikiArt, ResNet-50) to compute ASR, and standard metrics (FID, CLIP score) with MSCOCO for generation quality. Randomness is controlled by using multiple seeds per prompt in dataset construction.

**Compute.** Experiments were run on a single NVIDIA A40 GPU. We report the average required time to attack each data point in the main paper.

**Baselines.** We evaluate against UnlearnDiff, CCE, RECE, and other baselines using their public implementations and settings to ensure fair comparisons.

We believe these details, along with the planned public release of code and data splits, will enable full reproduction of our results.

## 9 USE OF LLMS

Large language models (LLMs), including ChatGPT and Google Gemini, were used solely to assist in editing and polishing the writing of this paper. All research ideas, experiments, and analyses were conducted independently by the authors.

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

## A  RELATED WORKS

**T2I Diffusion Models and Machine Unlearning.**  Text-to-image (T2I) diffusion models Rombach et al. (2022); Chang et al. (2023); Luo et al. (2023); Saharia et al. (2022); Gafni et al. (2022); Ramesh et al. (2022b); Yu et al. (2022); Xu et al. (2024) can take prompts as input and generate desired images following the prompt. There are several different types of T2I models, such as stable diffusion Rombach et al. (2022), latent consistency model Luo et al. (2023), and DeepFloyd Saharia et al. (2022). Despite their generation ability, safety concerns arise since these models have also gained the ability to generate unwanted images that are harmful or violate copyright. To solve this problem, some early works deploy safety filters Nichol et al. (2022); Rombach et al. (2022) or modified inference guidance Schramowski et al. (2023) but exhibit limited robustness Chin et al. (2024a); Yang et al. (2024). Recently, machine unlearning (MU) Nguyen et al. (2024); Ginart et al. (2019) is one of the major strategies that makes the model "forget" one specific concept via fine-tuning, and most MU works build on the widely used latent diffusion models (LDM), specifically stable diffusion (SD) models. Most diffusion machine unlearning works finetune the denoising UNets Gandikota et al. (2023); Zhang et al. (2023); Lyu et al. (2024); Kumari et al. (2023); Gandikota et al. (2024); Fan et al. (2024); Huang et al. (2024); Heng & Soh (2023). Although MU is a more practical solution than filtering datasets and retraining models from scratch, the robustness of MU still needs careful attention. Although current diffusion unlearning methods typically target the removal of a single concept per model, the need to preserve safe concept generation makes complete removal a challenging problem.

**Jailbreaking Attacks and Defenses on Unlearned Models.**  Recent works explore jailbreaking attacks on unlearned diffusion models, which aim to make unlearned models regenerate unwanted concepts. Such attacks can serve as a way to evaluate the robustness of unlearned diffusion models. For example, UnlearnDiff Zhang et al. (2024d) learns an adversarial attack prompt and appends the prompt before the original text prompt to do attacks, along a similar line of prior attack works Yang et al. (2023); Maus et al. (2023); Chin et al. (2024b); Tsai et al. (2024); Zhuang et al. (2023). Besides, the most related work to ours is Pham et al. (2024), utilizing Textual Inversion Gal et al. (2023). It also learns a token embedding that represents the target concept. Though we experimentally show CCE is in nature global to both text prompts and random noise as well, but is less transferable to different unlearned models. Prior jailbreaking attacks also do not consider the interpretability of the resulting attack prompts, thus offering limited insights into the underlying causes of the deficiencies in current unlearning methods, nor do they explore the potential for defense. In contrast, our attack token embeddings are interpretable and reveal the human-interpretable associations remained in unlearned diffusion models to "remember" the target concepts. Also, our method can be easily extended to learn a diverse set of attack token embeddings independent of each other. This diversity sheds light on the volume of the inner space where the target concept is still hidden. This motivates us to propose a simple yet effective defense method against existing attack methods. To the best of our knowledge, the defense of unlearned models is an underexplored problem in the field. A recent work, RECE Gong et al. (2024), targets a specific unlearned model (i.e., UCE Gandikota et al. (2024)), and focuses on defending it against adversarial attacks (i.e., UnlearnDiff). Defending a broader range of unlearned models against diverse attack types remains a challenging problem—one we aim to address by leveraging our defense.

**Diffusion Model Interpretability.**  To understand the semantics within diffusion models for applications such as image editing and decomposition, a series of works have attempted to interpret the representation space within diffusion models Kwon et al. (2023); Park et al. (2023b); Chen et al. (2024); Chefer et al. (2024). For example, Kwon et al. (2023) studies the semantic correspondences in the middle layer of the denoising UNet in diffusion models, while Chen et al. (2024) investigates the low-rank subspace spanned in the noise space. Some works Hertz et al. (2023); Han et al. (2023) focus on the visualization of attention maps with respect to input texts, while other works study the generalization and memorization perspective of diffusion models Zhang et al. (2024a). The most related work to ours is Chefer et al. (2024), which decomposes a single concept as a combination of a weighted combination of interpretable elements, in line with the concept decomposition and visualization works in a wider domain Olah et al. (2017); FEL et al. (2023); Bau et al. (2017). Inspired by Chefer et al. (2024) as well as other prior works, we attack unlearned diffusion models by learning

interpretable representations, which leads to further investigation on the root of failures for existing unlearned diffusion models, as well as a defense method.

**Linear Representation Hypothesis.**  In large language models (LLMs), the linear representation hypothesis posits that certain features and concepts learned by LLMs are encoded as linear vectors in their high-dimensional embedding spaces. This is supported by the fact that adding or subtracting specific vectors can manipulate a sentence's sentiment or extract specific semantic meanings Park et al. (2023a). The linear property has been further explored for understanding, detoxing, and controlling the generation of LLMs Liu et al. (2024). Similarly, other works investigating the representations of multimodal models find that concepts are encoded additively Radford et al. (2021); Yuksekgonul et al. (2023), and concepts can be decomposed by human-interpretable words Bhalla et al. (2024). Moreover, in stable diffusion models, Chefer et al. (2024) finds that concepts can be decomposed in the CLIP token embedding space in a bag-of-words manner. Based on these works, and considering the flexibility of the token embedding space in diffusion personalization Gal et al. (2023) and attacking Pham et al. (2024), we specifically investigate interpretable jailbreaking attacks and defenses for diffusion model unlearning by learning an attack token embedding that is a linear combination of existing token embeddings.

## B  EXPERIMENT SETTINGS

### B.1  ATTACK

**Unlearned LDMs as Victim Models.**  The field of diffusion unlearning is evolving rapidly, and there is a wide range of unlearning methods, most of which finetune the stable diffusion model. Most of the existing methods focus on single-concept unlearning. Following the protocol of Zhang et al. (2024d), we select several unlearned diffusion models that have an open-source and reproducible codebase, reasonable unlearning performance, and reasonable generation quality. This selection includes three widely used models from prior jailbreaking studies, namely ESD Gandikota et al. (2023), FMN Zhang et al. (2023), and UCE Gandikota et al. (2024), as well as more recent or complementary settings such as SPM (Lyu et al., 2024), MACE (Lu et al., 2024), SA (Heng & Soh, 2023), AC (Kumari et al., 2023), SalUn (Fan et al., 2023), and EraseDiff (Wu et al., 2024). These methods fine-tune the denoising UNet for unlearning while freezing other components. In our study, the unlearned models are fine-tuned on Stable Diffusion v1.4, and hence, they share the same CLIP text encoders.

**Attacking Dataset.**  Our learned token embedding represents the target concept, so the attack token embedding in nature can attack the victim model with different initial noise and text prompts. Thus, we construct a dataset to test such global attacking ability. To facilitate reproducibility, we follow the dataset construction protocol of UnlearnDiff as follows. We study three kinds of target concepts: "nudity" for NSFW, "Van Gogh" for artistic styles, and "church", "garbage truck", "parachute", and "tench" for objects. For each of "nudity", "Van Gogh", and "church", we prepare a corresponding dataset containing 900 (prompt, seed) pairs, and mainly use these concepts for baseline comparisons with other attacks. For each of the other concepts, we prepare a dataset of size 300. Each prompt contains the target concept to attack - for instance, "a photo of a nude woman in a sunlit garden" is an example prompt in the "nudity" dataset. Each prompt is associated with 10 - 30 different random seeds controlling the initial noise, and this results in a total of 300 - 900 (prompt, seed) pairs for each concept. Each pair is verified to produce the target concept with the original SD v1.4. Our dataset is approximately six times larger than that used in UnlearnDiff, enabling more reliable evaluation.

**Learning Details.**  We use SD 1.4 to generate 100 images containing the target concept as the training image dataset. The prompt used to generate images for each concept is similar to "A photo of a [target concept]". After that, to optimize each of the attack token embeddings for conducting SubAttack, we train an MLP network using the AdamW optimizer for 500 epochs with a batch size of 6. The MLP consists of two linear layers with ReLU activation applied after each layer. The first layer maps from 768 to 100 dimensions, and the second maps from 100 to 1. Experimental results confirm that this design has sufficient capacity to learn the scalar $\alpha_i$ for each embedding in the vocabulary. All experiments are conducted on a single NVIDIA A40 GPU.

**Attacking Details.** For NoAttack, the original text prompts and seeds are passed to the victim model. In SubAttack and CCE attacks, we replace the target concept in the text prompt with the special token associated with the learned attack token embedding (For example, change "a photo of a nude woman" to "a photo of a <$v_{att}$>"). In UnlearnDiff, we modify each text prompt by appending the corresponding learned adversarial prompt before it. For each attacking method and each concept, we generate 300-900 images using the resulting (prompt, seed) pairs for testing attack performance.

**Evaluation Protocols.** (*i*) After image generation, we use pretrained classifiers to detect the percentage of images containing the target concept following UnlearnDiff, and report it as the attacking success rate (ASR). For nudity, we use NudeNet Zhang et al. (2024d) to detect the existence of nudity subjects. For Van Gogh, we deploy the style classifier finetuned on the WikiArt dataset and released by Zhang et al. (2024d). We report the Top-3 ASR for style, i.e., if Van Gogh is predicted within the Top-3 style classes for a generated image, the image is viewed as a successful attack for Van Gogh style. For church, the object classifier pretrained on ImageNet Deng et al. (2009) using the ResNet-50 He et al. (2015) architecture is utilized. (*ii*) To evaluate the efficiency of different attack methods, we measure the average attack time required per image, which includes both the optimization time for learning embeddings or prompts and the generation time for creating images. For a given target concept dataset, CCE learns a single token embedding shared across all images and performs one generation per image. By default, SubAttack learns five shared token embeddings and generates five images per input. In contrast, UnlearnDiff performs up to 999 optimization iterations per image, requiring one image generation per iteration. As a result, UnlearnDiff is significantly more time-consuming than both CCE and SubAttack.

## B.2 DEFENSE

**Basics.** We follow the defending strategy presented in Sec. 3.2 by blocking a list of token embeddings for the entire CLIP vocabulary. SubDefens is plugged into UCE, ESD, FMN, and SPM. Defense performance is mainly assessed on concepts "nudity", "Van Gogh", and "church" using our constructed dataset. RECE, which defends UCE against UnlearnDiff, serves as the defending baseline and is compared with UCE+SubDefense with 20 blocked tokens. By default, in other cases, SubDefense is performed by learning and blocking 100 token embeddings. Both before and after cleaning up the token embedding space, we conduct independent attacks following the setting in App. B.1.

**Metrics.** An effective defense strategy should reduce the attack success rate while preserving the generation quality of safe concepts. Hence, we use the following metrics. (*i*) ASR. Various jailbreaking attacks are conducted before and after applying defenses, and the corresponding ASR is reported. Specifically for SubAttack, $K = 5$ is used consistently before and after defense to ensure a fair comparison. (*ii*) CLIP Score and FID are evaluated to test the generation quality of the defended model. MSCOCO Lin et al. (2014) contains image and text caption pairs. Following Zhang et al. (2024d;c), we use 10k MSCOCO text captions to generate images before and after defense. Then, we report the mean CLIP score Hessel et al. (2021) of generated images with their corresponding text captions to test the defended models' ability to follow these harmless prompts. And we report the FID between generated images and original MSCOCO images to test the quality of generated images.

## C AUXILIARY ATTACK RESULTS

### C.1 MORE INTERPRETATION RESULTS ON ATTACK TOKEN EMBEDDINGS

First of all, we show detailed results of transferring token embeddings from unlearned models to the original SD in **Tab. 6**, emphasizing that these embeddings are inherited from the original SD.

Moreover, we should provide additional interpretation of the sets of learned attack token embeddings for "church" and "Van Gogh" across different unlearned LDMs in **Fig. 10** and **Fig. 11**, showing observations on **interpretable associations** similar to that of "nudity".

For example, for "church", ESD (Fig. 10b) and UCE (Fig. 10d) majorly relate it with **religious concepts**, including names ("mary"), places ("abbey", "abby", "rom" for "rome"), etc. Interestingly, in **Scotland and Northern England English**, "kirk" is the traditional word for "church" - this may

Table 6: **Token embeddings learned by SubAttack originate from the original SD.** This is evidenced by the successful transfer of attack token embeddings from unlearned models to the original SD with high ASR.

| Scenarios: | ESD→SD | FMN→SD | UCE→SD | SPM→SD |
|---|---|---|---|---|
| Nudity | 97.44% | 97.78% | 95.89% | 86.11% |
| Van Gogh | 86% | 84% | 88.44% | 93.11% |
| Church | 87.22% | 92.56% | 85.56% | 84.33% |

be integrated into LDM during the training of large-scale datasets, but not removed during existing diffusion unlearning methods. As for FMN (Fig. 10c) and SPM (Fig. 10e), the **explicit concept "church"** itself is a significant component. Notably, FMN and SPM also exhibit higher ASR with no attack as presented in Fig. 3 and Tab. 7. Under NoAttack, both of them achieve ASR greater than 40%, but ASR for ESD and UCE is less than 10%. This also emphasizes that explicit associations also remain in some unlearned LDMs.

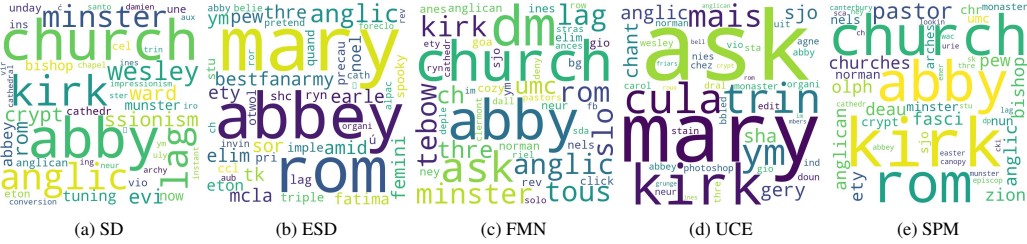

(a) SD     (b) ESD     (c) FMN     (d) UCE     (e) SPM

Figure 10: Interpreting attack token embeddings for the concept "church".

As for the concept "Van Gogh", when interpreting the sets of embeddings collectively, more **explicit words** are exposed for existing unlearned models such as "vincent", "gogh", "vangogh", along with **implicit words** "art", "artist", "munch" (Edvard Munch is an impressionist sharing similar themes and styles with Van Gogh, and the Van Gogh Museum in Amsterdam and the Munch Museum have collaborated to give a joint exhibition, "Munch: Van Gogh".) "monet" (also an impressionist), "nighter" and "oats" (concepts commonly in Van Gogh's paintings), etc. Although UCE, which shows the highest ASR with no attack, has the largest amount of explicitly associated concepts, other unlearned models all show explicit words more or less. This suggests that current unlearning methods retain more explicit associations with the target concept when applied to styles, compared to their application to NSFW and object concepts.

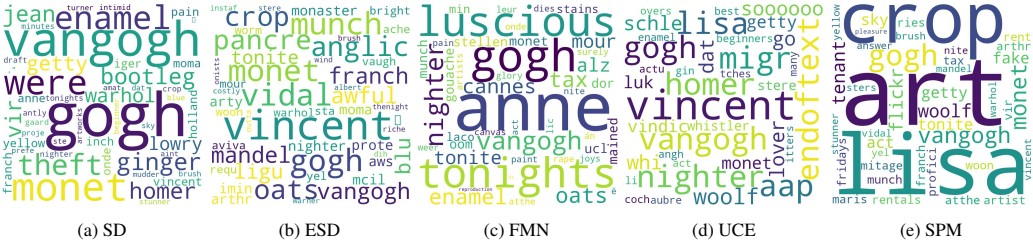

(a) SD     (b) ESD     (c) FMN     (d) UCE     (e) SPM

Figure 11: Interpreting attack token embeddings for the concept "Van Gogh".

## C.2 MORE ASR RESULTS

We present SubAttack ASR details with K=5 on different models across six concepts in **Tab. 7**. Moreover, we show ASR on a broader range of unlearned LDMs and settings such as massive concepts in **Tab. 8**, **Tab. 9**, and **Tab. 10**. Further more, we show transfer attack performance details from ESD to other unlearned models using different attack methods across different concepts in **Tab. 11**, **Tab. 12**, and **Tab. 13**. Moreover, we present additional transfer results between other unlearned model pairs using SubAttack with K=5 in **Tab. 14**.

Table 7: **Attack success rates (ASR)** targeting different unlearned diffusion models across different concept unlearning tasks (NSFW, artist style, object).

| Attacks: | NoAttack | | | | Ours | | | |
|---|---|---|---|---|---|---|---|---|
| **Victim Model:** | ESD | FMN | UCE | SPM | ESD | FMN | UCE | SPM |
| Nudity | 18.78% | 90% | 23% | 22.56% | 97.56% | 100.00% | 81.67% | 74.89% |
| Van Gogh | 5.78% | 21.56% | 71.44% | 43.78% | 81% | 96.33% | 98.33% | 82.78% |
| Church | 9.33% | 51.56% | 6.55% | 43.78% | 91.33% | 97.78% | 82.67% | 84.89% |
| Garbage Truck | 4% | 41.33% | 11.33% | 12.67% | 31.33% | 91.67% | 44% | 77.67% |
| Parachute | 4% | 63.67% | 1.3% | 30.67% | 88.67% | 100% | 67% | 97% |
| Tench | 1.67% | 40% | 0% | 14.33% | 26.67% | 80% | 49% | 84.33% |

Table 8: **Evaluation across diverse concepts and settings including MACE, SA, and AC.**

| Scenarios: | MACE (Nudity) | MACE (Truck) | MACE (Airplane) | MACE (Ship) | SA (Nudity) | AC (Van Gogh) |
|---|---|---|---|---|---|---|
| NoAttack | 6.67% | 10% | 0% | 6.67% | 83.33% | 21.67% |
| SubAttack (Ours) | **98.33%** | **85.56%** | **96.67%** | **100%** | **98.33%** | **61.67%** |

Table 9: **Attack success rates (ASR) against additional unlearned models including SalUn and EraseDiff.**

| Scenarios: | Church | Garbage Truck | Parachute | Tench |
|---|---|---|---|---|
| SalUn (NoAttack) | 1.67% | 5% | 5% | 0% |
| SalUn (SubAttack) | **56.67%** | **40%** | **86.67%** | **11.67%** |
| EraseDiff (NoAttack) | 6.67% | 6.67% | 3.33% | 0% |
| EraseDiff (SubAttack) | **31.67%** | **38.33%** | **78.33%** | **15%** |

Table 10: **Attack success rates (ASR) against RECE.**

| Scenarios: | Nudity | Van Gogh | Church |
|---|---|---|---|
| NoAttack | 3.33% | 16.67% | 3.33% |
| SubAttack | **62.44%** | **84.44%** | **80.33%** |

Table 11: **Transfer attack success rate for the concept "Nudity" using different attack methods.**

| Scenarios: | ESD→FMN | ESD→UCE | ESD→SPM |
|---|---|---|---|
| NoAttack | 90% | 23% | 22.56% |
| UnlearnDiff | 93.33% | 41.33% | 38.22% |
| CCE | 93% | 18.33% | 37.56% |
| SubAttack (Ours) | **96.89%** | **77%** | **80.44%** |

Table 12: **Transfer attack success rate for the concept "Van Gogh" using different attack methods.**

| Scenarios: | ESD→FMN | ESD→UCE | ESD→SPM |
|---|---|---|---|
| NoAttack | 21.56% | 71.44% | 43.78% |
| UnlearnDiff | 12.78% | 64% | 47.11% |
| CCE | 72.33% | 43.56% | 81.33% |
| SubAttack (Ours) | **72.67%** | **88.89%** | **86.89%** |

Table 13: **Transfer attack success rate for the concept "Church" using different attack methods.**

| Scenarios: | ESD→FMN | ESD→UCE | ESD→SPM |
|---|---|---|---|
| NoAttack | 51.56% | 6.55% | 43.78% |
| UnlearnDiff | 6.19% | 13.33% | 58% |
| CCE | 91% | 70.11% | **92.78%** |
| SubAttack (Ours) | **92.89%** | **83.77%** | 92% |

Table 14: **More SubAttack transfer results across four model pairs.**

| Scenario: | FMN->UCE | UCE->ESD | SPM->UCE | UCE->FMN |
|---|---|---|---|---|
| Nudity | 72% | 81.33% | 86.11% | 93.44% |
| Van Gogh | 91.11% | 48.55% | 80.55% | 62.55% |
| Church | 79.33% | 42.44% | 68.33% | 78.77% |

# D    AUXILIARY DEFENSE RESULTS

## D.1    DETAILED BASELINE COMPARISON OF DEFENDING UCE AGAINST UNLEARNDIFF

A more detailed comparison results of RECE and SubDefense together with UCE with no defense are presented in **Tab. 15** and **Tab. 16**.

Table 15: **SubDefense is stronger than baseline RECE in defending three concepts on UCE against UnlearnDiff or our SubAttack.**

| Attacks: | UnlearnDiff | | | SubAttack | | |
|---|---|---|---|---|---|---|
| Scenarios: | UCE | UCE + SubDefense | RECE | UCE | UCE + SubDefense | RECE |
| Nudity | 78.22% | 73.55% (**-4.67%**) | 76.44% (-1.78%) | 81.67% | 34.11% (**-47.56%**) | 62.44% (-19.23%) |
| Van Gogh | 100% | 52.78% (**-47.22%**) | 61.67% (-38.33%) | 98.33% | 29.44% (**-68.89%**) | 84.44% (-13.89%) |
| Church | 61.67% | 39.78% (**-64.34%**) | 50.78% (-10.89%) | 82.67% | 5.22% (**-77.45%**) | 80.33% (-2.34%) |

Table 16: **SubDefense preserves better utility than baseline RECE after defense.**

| Metrics: | COCO-10k FID (↓) | | | COCO-10k CLIP (↑) | | |
|---|---|---|---|---|---|---|
| Scenarios: | UCE | UCE + SubDefense | RECE | UCE | UCE + SubDefense | RECE |
| Nudity | **17.14** | 17.51 | 17.57 | **30.86** | 30.70 | 30.07 |
| Van Gogh | **16.64** | **16.64** | 17.11 | **31.14** | 30.94 | 30.08 |
| Church | 17.84 | **17.41** | **17.41** | **30.95** | 30.86 | 30.07 |

## D.2    DEFENDING AGAINST UNLEARNDIFF ON THE I2P DATASET FOR VARIOUS UNLEARNED MODELS

We construct dataset for concepts belonging to the style and object class following UnlearnDiff but with a larger size. Hence, defending against UnlearnDiff using these datasets can demonstrate the effectiveness of SubDefense in a scenario consistent with UnlearnDiff. However, for NSFW concepts such as nudity, UnlearnDiff filters prompts and seeds from the I2P dataset. Hence, to further test SubDefense's ability in defending against UnlearnDiff in this specific setting, we conduct UnlearnDiff with or without SubDefense using the I2P dataset as well. We report the defense results on ESD, FMN, UCE, and SPM in **Tab. 17**, **Tab. 18**, **Tab. 19**, and **Tab. 20** accordingly. We can see that SubDefense can reduce ASR on I2P consistently for all four models.

Table 17: **SubDefense for I2P-nudity on ESD against UnlearnDiff**, with 100 blocked tokens.

| Scenario: | ESD | ESD + SubDefense |
|---|---|---|
| NoAttack | 20.56% | 9.93% (-10.63%) |
| UnlearnDiff | 74.47% | 41.13% (-33.34%) |

Table 18: **SubDefense for I2P-nudity on FMN against UnlearnDiff**, with 100 blocked tokens.

| Scenario: | FMN | FMN + SubDefense |
|---|---|---|
| NoAttack | 87.94% | 37.59% (-50.35%) |
| UnlearnDiff | 97.87% | 45.39% (-52.58%) |

Table 19: **SubDefense for I2P-nudity on UCE against UnlearnDiff**, with 100 blocked tokens.

| Scenario: | UCE | UCE + SubDefense |
|---|---|---|
| NoAttack | 21.98% | 13.47% (-8.51%) |
| UnlearnDiff | 78.72% | 45.39% (-33.33%) |

Table 20: **SubDefense for I2P-nudity on SPM against UnlearnDiff**, with 100 blocked tokens.

| Scenario: | SPM | SPM + SubDefense |
|---|---|---|
| NoAttack | 55.31 % | 34.04% (-21.27%) |
| UnlearnDiff | 91.49 % | 58.97% (-32.52%) |

## D.3 DEFENDING AGAINST SUBATTACK ON VARIOUS CONCEPTS FOR VARIOUS UNLEARNED MODELS

Apart from the major baseline comparison of defense on UCE, and the defense results against different attacks on ESD presented in the main paper, we provide additional defense results of various concepts and unlearned models against SubAttack in this section. The results are shown in **Tab. 21**, **Tab. 22**, **Tab. 23**, and **Tab. 24** accordingly. Notice that ASR on various concepts is reduced with SubDefense, while ASR reduction on "Van Gogh" is the most significant. It is worth exploring in the future to design new methods and make the defense more robust for other concepts as well.

Table 21: **SubDefense for three concepts on ESD against SubAttack**, with 100 blocked tokens.

| Scenario: | ESD | ESD + SubDefense |
|---|---|---|
| Nudity | 97.56% | 42.33% (-55.23%) |
| Van Gogh | 81% | 17% (-64%) |
| Church | 91.33% | 40.22% (-51.11%) |

## D.4 DEFENDING RESULTS ON OTHER EXPLORATORY SETTINGS

**Defending against black-box attack.** We also conducted exploratory experiments on defending against Ring-A-Bell (Tsai et al., 2024), a classic black-box attack. Our results in **Tab. 25** show that SubDefense reduces ASR across several unlearned models, including MACE, FMN, SPM, and ESD. These findings suggest that SubDefense can provide robustness in black-box scenarios, although our main focus remains on white-box settings.

Table 22: **SubDefense for three concepts on FMN against SubAttack**, with 100 blocked tokens.

| Scenario: | FMN | FMN + SubDefense |
|---|---|---|
| Nudity | 100% | 62.89% (-37.11%) |
| Van Gogh | 96.33% | 22.78% (-73.55%) |
| Church | 82.67% | 13.78% (-68.89%) |

Table 23: **SubDefense for three concepts on UCE against SubAttack**, with 100 blocked tokens.

| Scenario: | UCE | UCE + SubDefense |
|---|---|---|
| Nudity | 81.67% | 28% (-53.67%) |
| Van Gogh | 93.78% | 14.33% (-79.45%) |
| Church | 82.67% | 3.22% (-79.45%) |

Table 24: **SubDefense for three concepts on SPM against SubAttack**, with 100 blocked tokens.

| Scenario: | SPM | SPM + SubDefense |
|---|---|---|
| Nudity | 74.89% | 50.78% (-24.11%) |
| Van Gogh | 82.78% | 12.33% (-70.45%) |
| Church | 84.89% | 23.78% (-61.11%) |

Table 25: **Exploratory defense results against the black-box Ring-A-Bell (Nudity) attack.**

| Scenarios: | MACE | FMN | SPM | ESD |
|---|---|---|---|---|
| Ring-A-Bell ASR | 11.58% | 95.79% | 34.74% | 57.89% |
| + SubDefense | **5.26%** (k=10) | **54.75%** (k=100) | **14.74%** (k=100) | **4.21%** (k=100) |

**Standalone performance of SubDefense.** Although SubDefense was primarily designed as a plug-in defense to enhance the robustness of existing unlearned models (similar to RECE operating on UCE), we also explored its effectiveness as a standalone unlearning method. Specifically, we applied SubDefense directly to the original Stable Diffusion (SD) model without any prior unlearning. Results are promising: as shown in **Tab. 26**, SubDefense reduces ASR under both black-box (Ring-A-Bell, Nudity) and white-box (SubAttack, Church) attacks.

Table 26: **Standalone performance of SubDefense.**

| Scenarios: | SD | K=10 | K=20 | K=50 | K=100 | K=150 | K=200 |
|---|---|---|---|---|---|---|---|
| Ring-A-Bell (Nudity) | 97.89% | 89.47% | 76.84% | 60% | 38.94% | 23.16% | 8.42% |
| SubAttack (Church) | 100% | 80% | 78.33% | 55% | 46.67% | 21.67% | 10% |

# E ABLATIONS

## E.1 ATTACK

**Number of attack tokens.** In practice, we use $K = 5$ to conduct SubAttack as it provides strong attack performance while maintaining computational efficiency. Here, we take ESD as an example to show how ASR varies with $K$. To conduct ablations more efficiently, we subsample 300 out of 900 prompts for the concepts "church" and "nudity" to study the relationship between ASR and $K$. Results are presented in **Fig. 12** and **Fig. 13**. The additional attack time per image caused by

each additional token embedding is approximately 10 seconds, which leads to about 3 more hours to attack a single concept having 900 prompts in the dataset. Therefore, considering the needs of attacking multiple concepts and multiple models in practice, we choose $K = 5$ where the ASR is approximately stabilized. For some unique scenarios, users can choose to increase $K$ for higher ASR at a cost of longer computation time.

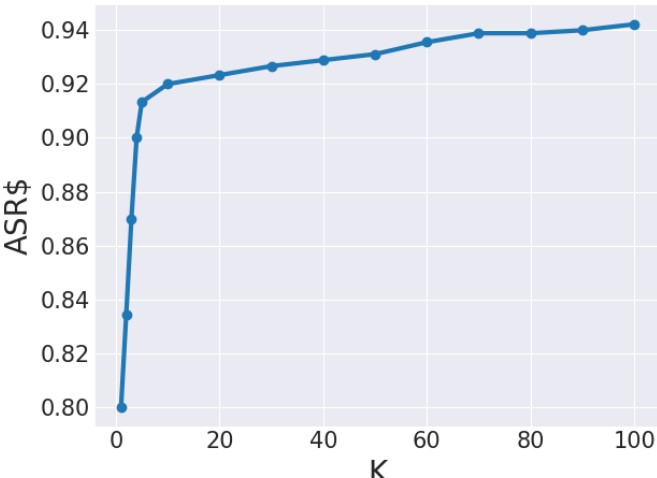

Figure 12: **ASR versus** $K$ when conducting SubAttack on ESD for the concept "church".

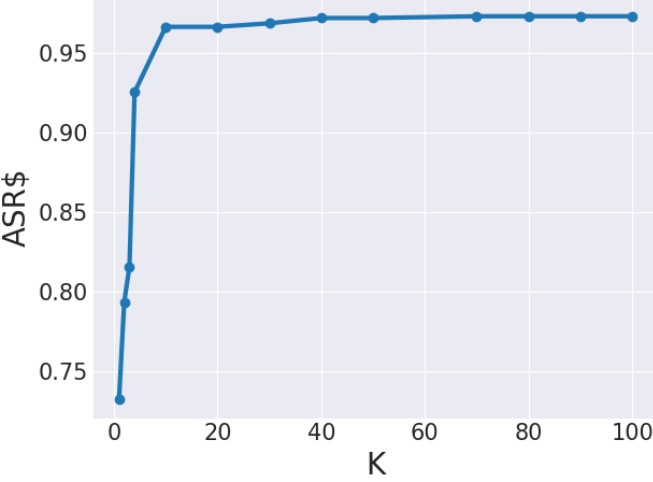

Figure 13: **ASR versus** $K$ when conducting SubAttack on ESD for the concept "nudity".

**Orthogonility.** The orthogonality constraint was introduced to encourage diversity among the learned attack embeddings, preventing them from collapsing into a single semantic direction and thereby covering a broader and more effective attack space. To validate this design choice, we conducted an ablation study on the "Nudity" concept. As shown in **Tab. 27**, enforcing orthogonality consistently improves ASR across multiple unlearned models, supporting the effectiveness of this constraint.

**Vocabulary size.** We ablate the vocabulary size used for SubAttack by selecting the top-N CLIP tokens most similar to the target concept. As shown in **Tab. 28**, ASR improves sharply up to 5000 tokens but declines when the vocabulary grows larger. This indicates that 5000 tokens strike the best balance between diversity and optimization feasibility.

Table 27: **Ablation on the orthogonality constraint.** Enforcing orthogonality improves ASR across unlearned models for the "Nudity" concept.

| Scenarios: | ESD | FMN | UCE | SPM |
|---|---|---|---|---|
| With Orthogonality | **97.56%** | **100%** | **81.67%** | **74.89%** |
| Without Orthogonality | 78.33% | 100% | 71.67% | 63.33% |

Table 28: **Ablation on vocabulary size.** ASR of SubAttack on ESD "Nudity" with different vocabulary sizes.

| Vocabulary size | 50 | 500 | 5000 (default) | 10000 | Full |
|---|---|---|---|---|---|
| ASR | 43.33% | 81.67% | **97.56%** | 70% | 28.33% |

### E.2 DEFENSE

**Gradual degradation of generation utility with stronger defense.** We show an ablation study on COCO-10k generation CLIP score and FID versus the number of blocked tokens in **Tab. 29** using ESD for the concept of "nudity". We can see that, after the number of blocked tokens surpasses 100, there appears to be a significant harm to the CLIP score and FID. In practice, the number of blocked tokens during defense can be selected to balance good generation quality and low ASR according to one's preference. In this paper, we provide an ablation study on ESD as an example, and report ASR majorly with 20 or 100 blocked tokens for different unlearned models and concepts.

Table 29: **SubDefense exhibits gradual degradation of CLIP score and FID when the number of blocked token embeddings increases.**

| #Blocked Tokens: | 0 | 20 | 50 | 100 | 200 | 300 | 350 |
|---|---|---|---|---|---|---|---|
| CLIP Score ($\uparrow$) | 30.13 | 30.02 | 29.86 | 29.58 | 28.54 | 26.15 | 24.72 |
| FID ($\downarrow$) | 18.23 | 19.02 | 19.09 | 19.20 | 20.92 | 26.42 | 30.33 |

**More results and discussions on defending against CCE.** Defending against CCE is an underexplored problem in the field, where there are no baselines to compare with, to the best of our knowledge. Hence, we show a detailed study on defense against CCE, along with more discussions to support future research. As shown in **Tab. 30**, different from UlearnDiff, CCE requires a large number of tokens to be blocked if we aim to have low ASR. However, lower ASR achieved by more blocked attack tokens leads to a degradation of generation utility, with an increased FID and a decreased CLIP score, referring to **Tab. 29**. Such a phenomenon indicates that the embedding identified by CCE has a complex association with the target concept, sharing components with a variety of interpretable token embeddings found by our method. This suggests that fully understanding the behavior of CCE requires a deeper analysis of how LDMs interpret and generate concepts other than the current approach we use. For example, currently, the interpretability of retained associations of concepts relies on predefined CLIP vocabularies, which may not capture all implicit or nuanced representations retained in unlearned models. While the above question is beyond the scope of the current work, such insights could inform the development of more robust and versatile defense strategies in the future. With improved understanding of LDMs, future research may come up with more efficient and robust defenses against CCE while preserving model utility.

Table 30: **ASR of concept "nudity" on CCE after blocking different numbers of token embeddings.**

| #Blocked Tokens: | 0 | 100 | 230 | 270 | 320 | 350 | 390 |
|---|---|---|---|---|---|---|---|
| CCE ASR | 85.11% | 75.67% | 65.78% | 37.44% | 28.11% | 18.11% | 8.89% |

# F   SPARSITY OF ATTACK TOKEN EMBEDDINGS

Sparsity constraints are widely adopted in prior concept decomposition works - where the linear combination coefficients $\alpha_i$ are forced to be nearly zeros except for dozens of tokens (usually 20-50). However, in our attacks, where the unlearned diffusion models majorly associate the target concept with a set of implicit tokens, removing such sparsity regularization is helpful, especially for attack token embeddings discovered later in the iterative learning process. Hence, we do not impose a sparsity constraint. Yet, it's interesting to find through our learning that a weaker sparse structure still emerges, and such sparsity gradually decreases as we learn more attack token embeddings through the iterative learning process.

Specifically, for each learned attack token embedding, we normalize $\boldsymbol{\alpha} = [\alpha_1, \dots, \alpha_N]$ to have a unit norm. Then, we find the index $i^*$ such that:

$$i^* = \arg\min_i i, \quad \text{such that} \quad \sum_{j=1}^{i} \alpha_j^2 \geq 0.9 \tag{5}$$

Besides, we also count the number of $\alpha_i$ such that $\alpha_i \geq 0.01$. We report the results of the first attack token embedding on ESD for each concept in **Tab. 31**. Notice the size of the CLIP token vocabulary is more than 40000.

Table 31: **Sparsity of the learned attack token embeddings.**

| Concept: | Nudity | Van Gogh | Church |
|---|---|---|---|
| $i^*$ | 1455 | 668 | 547 |
| $\#\alpha_i \geq 0.01$ | 1743 | 1023 | 885 |

During our iterative learning process of a set of tokens for the nudity concept, we observe a decreasing sparsity, as shown in **Tab. 32**. This is intuitive since later attacking requires more complex associations to the target concept.

Table 32: **Sparsity of the learned attack token embeddings decreases during the iterative subspace attack process.**

| #Itrs | 1 | 10 | 30 | 50 | 70 | 100 | 130 | 150 | 170 | 200 |
|---|---|---|---|---|---|---|---|---|---|---|
| $i^*$ | 1455 | 1799 | 1905 | 1784 | 1914 | 2062 | 2062 | 2136 | 2155 | 2115 |
| $\#\alpha_i \geq 0.01$ | 1743 | 2019 | 2078 | 2009 | 2206 | 2298 | 2328 | 2368 | 2358 | 2326 |

Furthermore, we visualize the nudity concept attacking results on ESD by selecting only the largest dozens of $\alpha_i$ within a learned $\boldsymbol{\alpha}$ and setting other entries as zeros. As shown in **Fig. 14**, we see the nudity concept is gradually enhanced as the number of selected $\alpha_i$ increases to 1500: the woman generated happens to wear fewer and fewer clothes until she's completely bare.

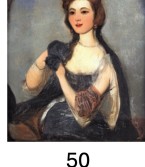 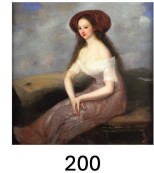 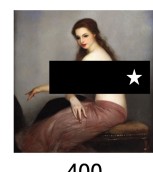 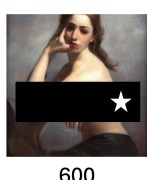 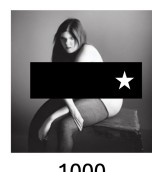 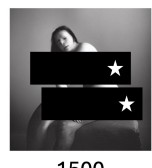

| 50 | 200 | 400 | 600 | 1000 | 1500 |

Figure 14: Attacking the concept nudity on ESD when $\boldsymbol{\alpha}$ has different numbers of non-zero entries.

## G    Image Generation Quality Visualization After Defense

In this section, we provide a more detailed study on the generation quality of unlearned models after we plug SubDefense into them. First, we provide more detailed MSCOCO prompts and the generated images of UCE and UCE + SubDefense (with 20 blocked tokens) in **Fig. 15**, **Fig. 16**, and **Fig. 17**. Next, taking UCE and "Van Gogh" as an example, whose attack token embeddings are highly related to "blue" and "star", we study whether SubDefense of "Van Gogh" harms the generation of "blue" and "star" in **Fig. 18** and **Fig. 19**. It turns out that the ability to generate these related concepts is highly preserved, which highlights that subdefense is different from direct token blocking of all related concepts. Instead, SubDefense blocks the composed embeddings, which represent the concept "Van Gogh" more accurately.

Figure 15: **More detailed visualization of COCO generation results with or without SubDefense on the concept nudity.**

Figure 16: **More detailed visualization of COCO generation results with or without SubDefense on the concept Van Gogh.**

Figure 17: **More detailed visualization of COCO generation results with or without SubDefense on the concept church.**

Figure 18: **Visualization of "blue" image generation results before and after defending "Van Gogh" on UCE.**

| Prompt | UCE | UCE + SubDefense |
|--------|-----|------------------|
| "Bright star in the night sky." | | |
| "Galaxy with many stars." | | |
| "Glowing star-shaped lantern." | | |

Figure 19: **Visualization of "star" image generation results before and after defending "Van Gogh" on UCE.**

## H MORE ATTACK VISUALIZATIONS

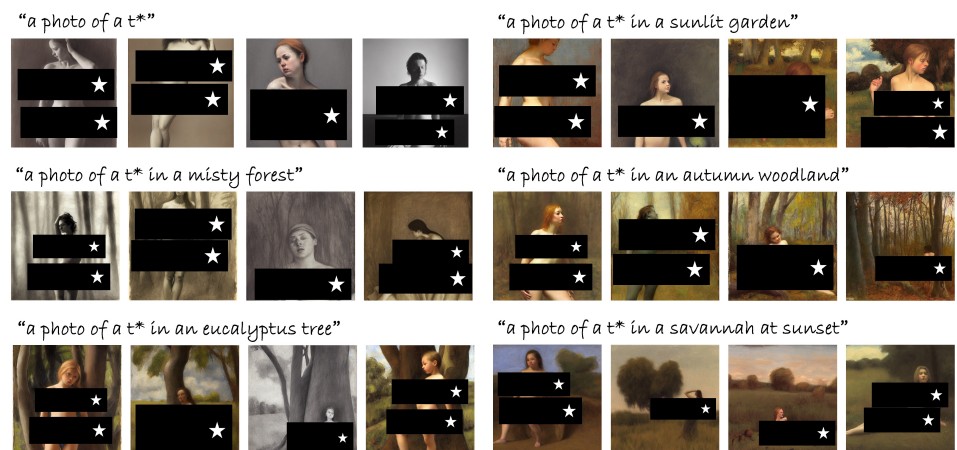

Figure 20: **Visualizing nudity attacking results on ESD.**

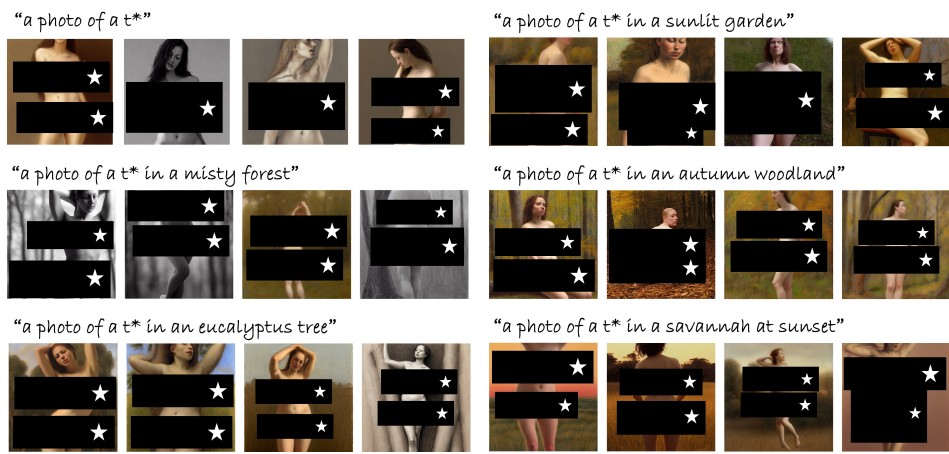

Figure 21: **Visualizing nudity attacking results on FMN.**

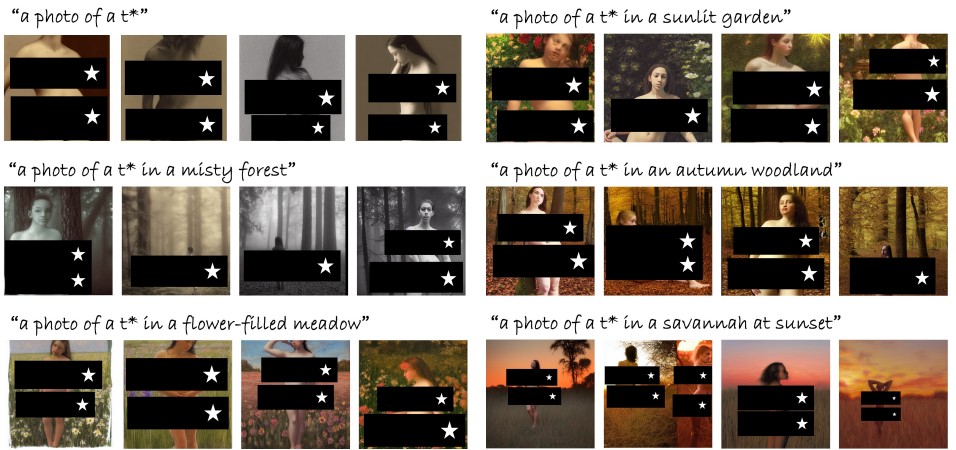

Figure 22: **Visualizing nudity attacking results on UCE.**

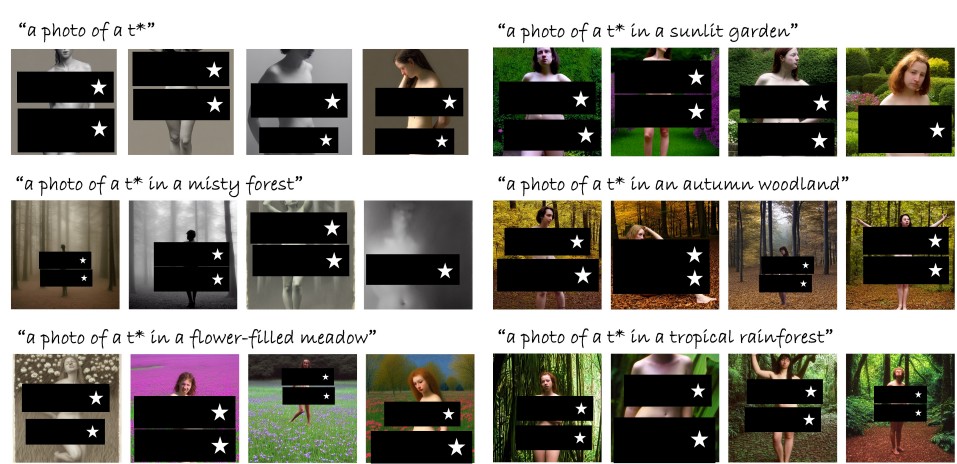

Figure 23: **Visualizing nudity attacking results on SPM.**

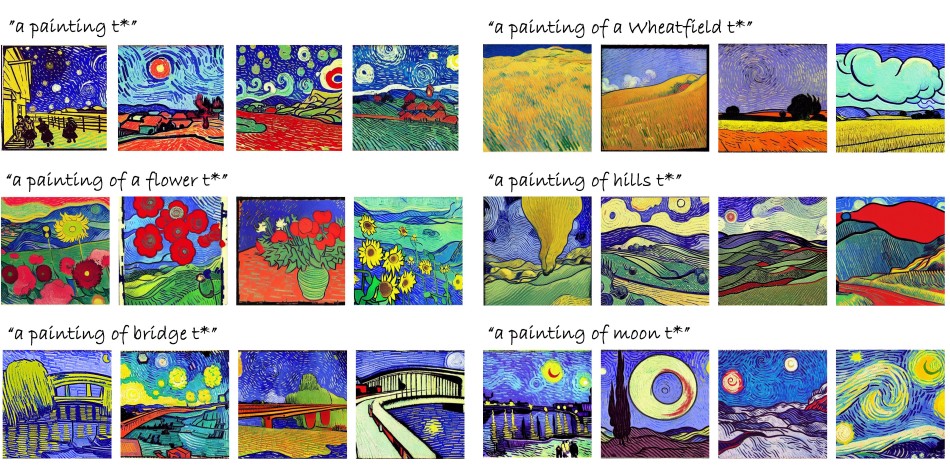

Figure 24: **Visualizing Van Gogh attacking results on ESD.**

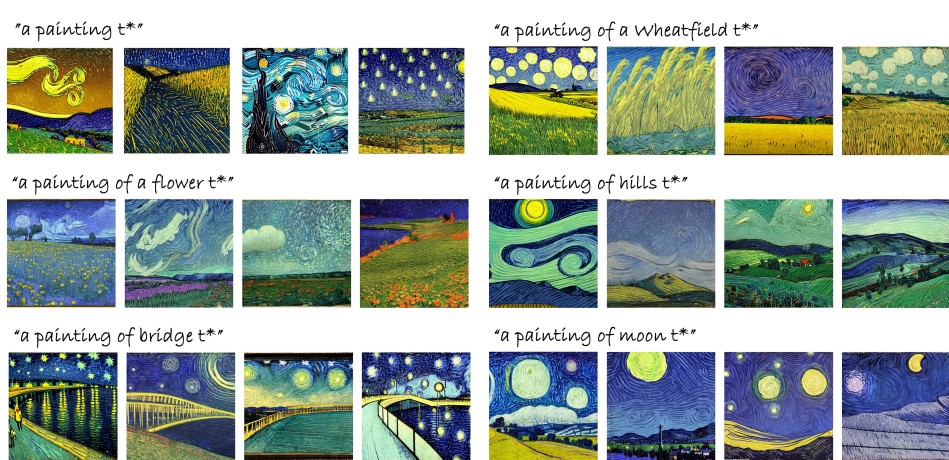

Figure 25: **Visualizing Van Gogh attacking results on FMN.**

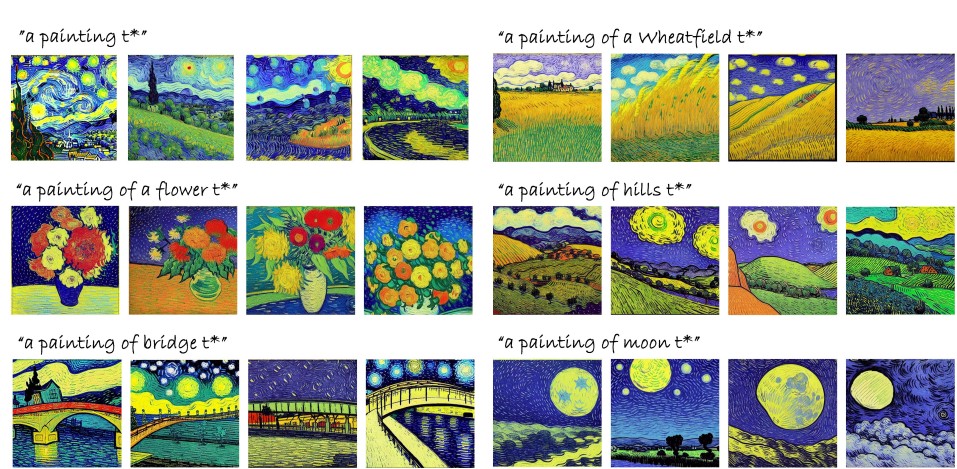

Figure 26: **Visualizing Van Gogh attacking results on UCE.**

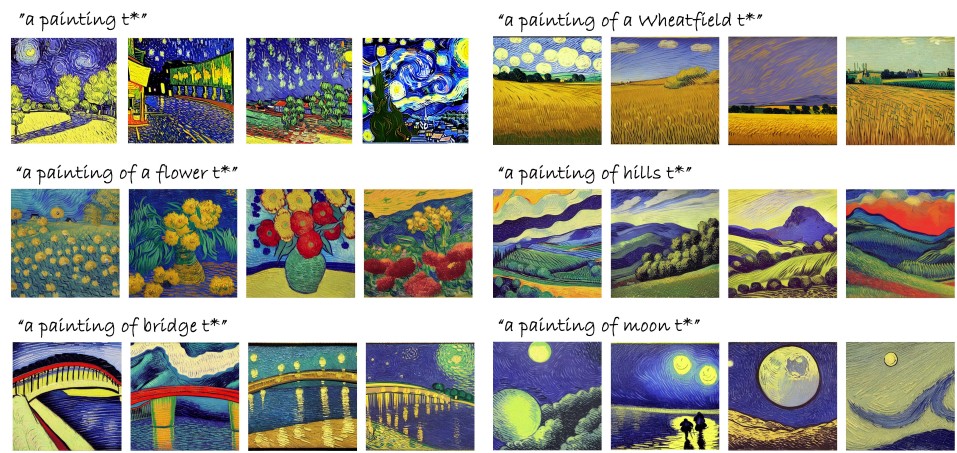

Figure 27: **Visualizing Van Gogh attacking results on SPM.**

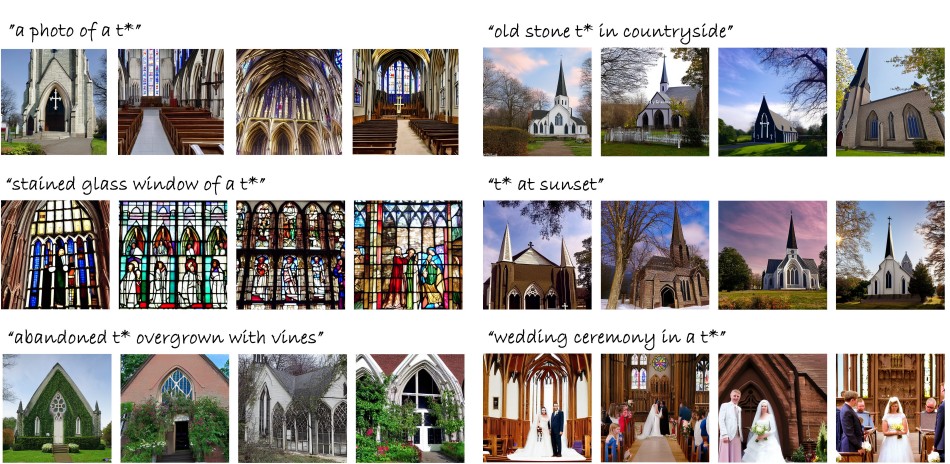

Figure 28: **Visualizing church attacking results on ESD.**

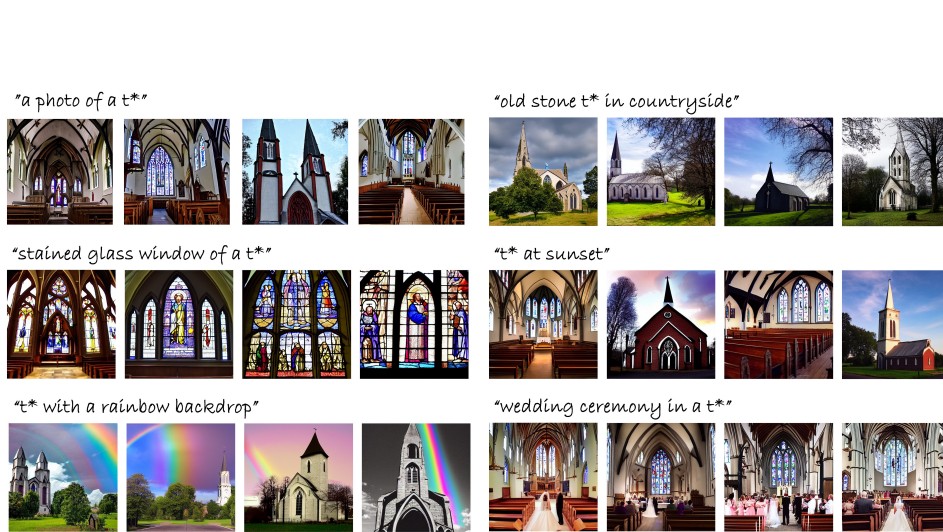

Figure 29: **Visualizing church attacking results on FMN.**

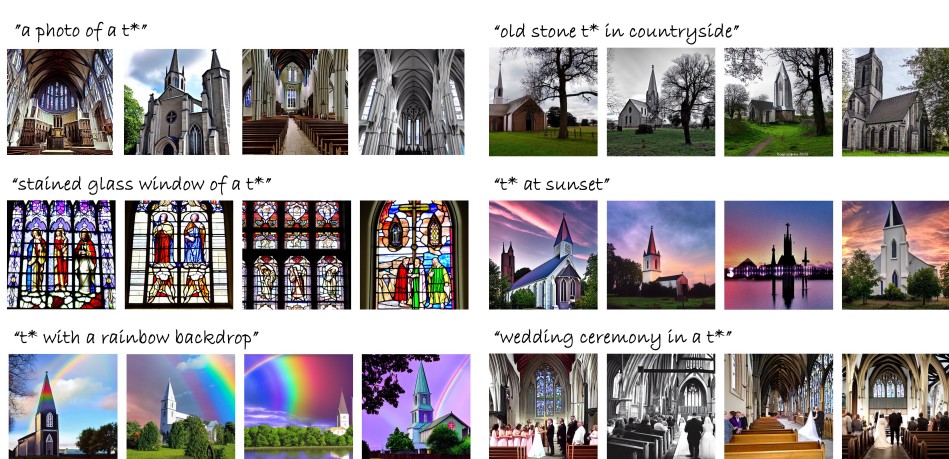

Figure 30: **Visualizing church attacking results on SPM.**

## I   FUTURE DIRECTIONS

We identify the following future directions. First, future research could explore feature representations in diffusion models beyond the linear structure, which may reveal richer mechanisms underlying unlearning. Second, efficient, adaptive, and automatic methods could be designed to determine not only the number of blocked tokens but also the specific set to block, for example through learned importance scores or attention-based relevance. Third, joint visual–textual embeddings could be investigated to better understand and defend against multimodal jailbreaks. Fourth, as a reference point for defenses against CCE, SubDefense highlights a clear trade-off between robustness and utility; addressing this trade-off remains an important open challenge. Fifth, extending SubDefense beyond CLIP-based architectures is another promising avenue. The core principle of identifying and nullifying harmful semantic directions in the conditional embedding space could be applied to other text encoders or even to models conditioned on alternative modalities. Finally, examining residual associations without relying solely on predefined vocabularies may capture more implicit or nuanced concepts retained in unlearned models, improving interpretability and guiding the development of stronger defenses.

