# OpenReview forum: "The Dual Power of Interpretable Token Embeddings: Jailbreaking Attacks and Defenses for Diffusion Model Unlearning"
_ICLR.cc/2026/Conference — Submitted to ICLR 2026_

### Official Review · Reviewer_76Nh · 2025-10-27

**Soundness:** 2
**Presentation:** 3
**Contribution:** 2
**Rating:** 2
**Confidence:** 4

**Summary:**

The authors propose two methods:

- SubAttack, a “human-interpretable” jailbreaking attack that constructs an orthogonal set of token embeddings via non-negative linear combinations of CLIP token embeddings;
- SubDefense, a defense mechanism that projects token embeddings orthogonally to remove residual concept subspaces.

The authors claim that SubAttack provides better interpretability, transferability, and efficiency than existing methods, and that SubDefense offers improved robustness with minimal degradation in generation quality.

**Strengths:**

The paper makes an attempt to interpret unlearned model vulnerabilities through embedding subspace analysis, which is important for safety and compliance of diffusion models.

**Weaknesses:**

1. Regarding the novelty of the method, although the concept of inserting v_att tokens is employed, this attack and defense strategy appears to essentially be a linear orthogonal transformation.
2. The method still requires training an MLP, which is not training-free, casting doubt on the stability of the attack strategy if there are many concepts that need to be attacked.
3. The scalability of the method is questionable, as restoring a single concept requires extensive training. This limits the generalizability of the attack approach.
4. The baselines compared in Figure 8,9 and Table 4 only include ECE and RECE, which seems to lack reliability.

**Questions:**

See weaknesses

**Details Of Ethics Concerns:**

No.

---

> ### Author Response · Authors · 2025-11-24
> **Author Response to Reviewer Feedback (1)**
>
> We thank the reviewer for the feedback and the opportunity to clarify several points. Below we address each concern in a concise and point-by-point manner.
>
> ---
>
> > ## 1. “The method is just a linear orthogonal transformation.”
>
>
> We appreciate the reviewer’s concern that the method might appear to be “just a linear orthogonal transformation.” However, we want to clarify two key points: **(1)** the linear assumption is highly motivated by extensive prior work on **linear interpretability** [1-7], and **(2)** crucially, the ***existence***, ***interpretability***, and further ***practical utility*** of **any linear structure** in **unlearned models** are **not guaranteed**, making it non-trivial to uncover such a structure or demonstrate its effectiveness for model **diagnosis** and **improvement**. We elaborate on these two points below.
>
> **(1)** Prior works in mechanistic interpretability [1-5] and the linear representation hypothesis [6-7] show that large models often encode semantic information in approximately linear latent spaces. Motivated by these findings, our aim is to examine **whether** the remaining concept in an unlearned diffusion model also organizes into a **coherent linear structure**, **what** this structure looks like, and **whether** it can be leveraged to systematically **diagnose** and **improve** the unlearned models.
>
> **(2)** Our findings show that the residual concept is **not arbitrary noise but human-interpretable** in a **linear subspace** that can be reliably identified using **non-negativity**, **sparsity**, and **orthogonality** constraints. This **linear interpretable structure** is what enables both our **diagnostic analysis** (SubAttack) and our **post-unlearning defense mechanism** (SubDefense).
>
> Concretely, our methods are:
>
> - **Interpretable structure discovery:** Section 4.2 shows that erased concepts persist along a coherent, human-interpretable **linear subspace**.
> - **Effective diagnosis:** **SubAttack** efficiently and effectively exposes remaining vulnerabilities with this linear subspace (Section 4.3).
> - **Effective improvement:** **SubDefense** removes these residual directions via a simple projection, enabling a lightweight post-unlearning refinement of unlearned models (Section 5.2).
>
> ---
>
> [1] L. Bereska and S. Gavves. *Mechanistic Interpretability for AI Safety – A Review*. TMLR, 2024.
>
> [2] Sharkey et al., *Open Problems in Mechanistic Interpretability*, TMLR 2025.
>
> [3] A. Zou et al. *Representation Engineering: A Top-Down Approach to AI Transparency*. arXiv:2310.01405, 2025.
>
> [4] Anthropic, *Scaling Monosemanticity: Extracting Interpretable Features from Claude 3 Sonnet*. 2024.
>
> [5] T. Bricken, S. Mishra-Sharma, S. Bhatt, K. Rivoire, et al. *Towards Monosemanticity: Decomposing Language Models With Dictionary Learning*. 2023.
>
> [6] K. Park, Y. J. Choe, and V. Veitch. *The linear representation hypothesis and the geometry of large language models*. ICML 2024.
>
> [7] M. Yüksekgönül, F. Bianchi, P. Kalluri, D. Jurafsky, and J. Zou. *When and Why Vision-Language Models Behave like Bags-of-Words, and What to Do About It?* ICLR 2023.

---

> ### Author Response · Authors · 2025-11-24
> **Author Response to Reviewer Feedback (2)**
>
> > ## 2. “The method still requires training an MLP, which is not training-free, casting doubt on the stability of the attack strategy for multi-concepts.”
>
> We appreciate the reviewer’s concern regarding the use of an MLP. We would like to clarify that we do not claim SubAttack is **training-free or unstable**. Rather, our claim is that the required training is **lightweight and stable** in practice.
>
> **Lightweight.**
> The MLP we train is intentionally **very small** and can be optimized to convergence within **minutes**. Moreover, the resulting attack embedding exhibits **strong transferability** across noise seeds, prompts, and even different unlearned models. This requires substantially less training than adversarial attacks that must re-optimize perturbations for **every** prompt, noise seed, or model instance.
>
> **Stable.**
> The ReLU-parameterized MLP naturally enforces **non-negativity**, which simplifies the optimization landscape and contributes to **stable** convergence.
>
> Together, these properties make SubAttack both **efficient** (Section 4.3(i)) and **effective** (Section 4.3(ii)):
>
> 1. It achieves the **highest ASR** in most evaluated settings (Table 3).
> 2. It is **far more efficient** than adversarial attacks requiring per-prompt and per-noise re-optimization (Table 3), taking only **54.2s** per attacking data point compared to **906.6s** for adversarial baselines.
> 3. It also scales well to **multi-concept** attack on the unlearned model MACE, as shown in **Appendix C.2** (Table 8), SubAttack achieves **98.33% / 85.56% / 96.67% / 100%** ASR across the four attacked concepts.
>
> ---
>
> > ## 3. “Scalability: restoring a single concept requires extensive training, limits the generalizability of the attack approach.”
>
> We appreciate the reviewer’s concern regarding scalability, and we would like to offer clarification along two dimensions.
>
> **(1) SubAttack requires far less training than adversarial attack baselines.**
> As discussed in **Q2 above**, SubAttack trains a **tiny, fast-converging MLP** and produces an attack embedding with **high transferability** across seeds, prompts, and even unlearned models.
> Because the training cost is very small and the learned embedding generalizes well, scaling to additional concepts is **substantially easier** for SubAttack than for adversarial attacks such as UnlearnDiff, which must be re-run for every new prompt/noise instance (**54.2s** per attacking data point for SubAttack compared to **906.6s** for unlearnDiff).
>
> **(2) The single-concept setting is the standard evaluation in unlearning, and we additionally provide multi-concept results.**
> Due to the intrinsic difficulty of balancing unlearning quality and model utility across many concepts, **almost all existing unlearning methods** focus on the **single-concept** setting. Our main paper follows this standard evaluation protocol.
> Nevertheless, we also present **multi-concept attack** results in **Appendix C.2** on the unlearned model MACE, which itself includes initial explorations into multi-concept unlearning. We show that SubAttack maintains strong performance in this multi-concept setting (Table 8, SubAttack achieves **98.33% / 85.56% / 96.67% / 100%** ASR across the four attacked concepts), further supporting its generalizability.

---

> ### Author Response · Authors · 2025-11-24
> **Author Response to Reviewer Feedback (3)**
>
> > ## 4. "The baselines only include UCE and RECE"
>
> ### **(1) Why UCE + RECE are the primary defense baselines**
> We appreciate the reviewer’s question regarding our choice of baselines. Our defense component, inspired by the interpretable diagnosis of unlearned models, focuses specifically on **post-unlearning refinement** rather than proposing a new adversarial unlearning algorithm (Section 2.2, *Defense*). SubDefense is designed to be a **plug-and-play refinement** that can be applied to *any* unlearned model.
>
> Within this regime:
>
> - **UCE** remains a widely used base unlearning model.
> - **RECE** is a prior method designed to **further refine the already-unlearned model, UCE,** without re-tuning a base model.
>
> Thus, the UCE/RECE comparison follows the same *refinement setting* and is chosen as the main baselines due to shared design and scope.
>
> ### **(2) Broader Generalization Across Models, Concepts, and Attack Scenarios**
> Due to space limits, the main paper highlights these baselines. **Appendix D** (Table 17–25) presents **broader defense evaluations**, including:
>
> - more unlearned **models** (SPM, ESD, FMN variants),
> - additional erased **concepts**, and
> - defending against **black-box** attacks (Ring-A-Bell).
>
> These results, with consistently and substantially reduced ASR (*e.g.*, **-52.58% / -68.89% / -90% ...**), demonstrate that SubDefense generalizes well in its intended plug-and-play role.
>
>
> ### **(3) Additional results**
>
> In addition, we now provide **SubDefense results on the recent strong unlearner STEREO**, showing that even a strong unlearned model can be *further improved* by SubDefense. For the concept *nudity*, SubDefense (with 20 blocked directions) reduces ASR substantially:
>
> | Model  | ASR Before Defense | ASR After Defense |
> |--------|----------------------|-------------------|
> | STEREO | **21.67%**           | **6.67%**         |
>
> These results reliably reinforce that our approach offers a simple and effective way to enhance unlearned models.
>
>
>
> ---
>
> We sincerely appreciate the reviewer’s time and thoughtful feedback, and we hope the above responses provide clear clarification.

---

> ### Comment · Reviewer_76Nh · 2025-11-28
>
> Thanks for your rebuttal. I appreciate the authors’ efforts in addressing my concerns. However, it appears that the system currently does not allow me to modify my score. If it becomes possible later, I will update my score.

---

### Official Review · Reviewer_SBEJ · 2025-10-28

**Soundness:** 3
**Presentation:** 3
**Contribution:** 3
**Rating:** 6
**Confidence:** 2

**Summary:**

This paper addresses a critical vulnerability in ​​diffusion model unlearning​​—where models fine-tuned to "forget" harmful concepts (e.g., nudity, copyrighted styles) remain susceptible to ​​jailbreak attacks​​ that regenerate erased content.

**Strengths:**

1. The proposed method is well motivated and demonstrates a strong level of interpretability, making the overall approach conceptually appealing.

2. The methodology is described with clarity, and the paper is easy to follow in terms of structure and technical presentation.

3. The experimental section is thorough and comprehensive.

**Weaknesses:**

1. The description of the authors’ main contributions in the methodology section is not entirely clear. If Section 3.1 corresponds to a prior baseline method and Section 3.2 presents the proposed SubAttack method, it appears that SubAttack might be an iterative extension or multi-token variant of the previous approach rather than a fundamentally new contribution. This raises some concerns regarding the novelty and the strength of the claimed contribution. I would appreciate clarification on this point in the rebuttal.

2. The evaluation of SubDefense appears limited in scope. The paper compares SubDefense with only a few baseline defense methods and under a relatively small number of attack scenarios. This experimental coverage may not be sufficient to convincingly demonstrate the superiority or generalizability of SubDefense.

**Questions:**

In the methodology section, the paper mentions that SubDefense is designed to defend against the CCE attack method. Does SubDefense require prior knowledge of the attack vector $v_{att}$ used by the adversary? If so, this assumption might be unrealistic in practical settings, and it would be helpful if the authors could clarify how such information is obtained or approximated.

---

> ### Author Response · Authors · 2025-11-24
> **Author Response to Reviewer Feedback (1)**
>
> We thank the reviewer for the constructive feedback and positive remarks. Below we address the three questions raised and have updated the PDF accordingly.
>
> ---
>
> > ## 1. Clarifying Contributions
>
> Thank you for the thoughful question. Section 3.1 summarizes a learning step inspired by related work, with constraints and the optimization procedure adapted specifically for conducting an attack. This is analogous to how CCE draws inspiration from Textual Inversion in image editing and adapts that idea for jailbreak attacks. We explicitly discuss these inspirations to clearly situate our method within prior work.
>
> SubAttack, however, is **not** a multi-token extension of existing single-vector attacks. Its contributions lie in:
> - **learning a subspace** of token directions using **orthogonality** and **auto-adapted sparsity** constraints;
> - uncovering **new interpretability findings unique to unlearned models**; and
> - motivating **SubDefense**, which removes these residual directions in a plug-and-play manner to refine unlearned models.
>
> The **existence** of such a linear subspace, its **interpretability and empirical implications** in unlearned models, and its further **utility** for both diagnostic attack and downstream defense are **non-obvious** and not straightforward extensions of prior work.
>
>
>
> ---
>
> > ## 2. SubDefense Evaluation
>
> Thank you for raising this concern. We provide below a structured clarification of our SubDefense evaluation and how our broader experiments demonstrate its generalizability.
>
> ### **(1) UCE + RECE are the primary defense baselines**
> We appreciate the reviewer’s question regarding our choice of baselines. Our defense component, inspired by the interpretable diagnosis of unlearned models, focuses specifically on **post-unlearning refinement** rather than proposing a new adversarial unlearning algorithm (Section 2.2, *Defense*). SubDefense is designed to be a **plug-and-play refinement** that can be applied to *any* unlearned model.
>
> Within this regime:
>
> - **UCE** remains a widely used base unlearning model.
> - **RECE** is a prior method designed to **further refine the already-unlearned model, UCE,** without re-tuning a base model.
>
> Thus, the UCE/RECE comparison follows the same *refinement setting* and is chosen as the main baselines due to shared design and scope.
>
> ### **(2) Broader Generalization Across Models, Concepts, and Attack Scenarios**
> Due to space limits, the main paper highlights these baselines. **Appendix D** (Table 17–25) presents **broader defense evaluations**, including:
>
> - more unlearned models (SPM, ESD, FMN variants),
> - additional erased concepts, and
> - defending against black-box attacks (Ring-A-Bell).
>
> These results show that SubDefense generalizes well in its intended plug-and-play role.
>
>
> ### **(3) Additional results**
>
> In addition, we now provide **SubDefense results on STEREO**, showing that even a strong unlearned model can be *further improved* by SubDefense. For the concept *nudity*, SubDefense (with 20 blocked directions) reduces ASR substantially:
>
> | Model  | ASR Before Defense | ASR After Defense |
> |--------|----------------------|-------------------|
> | STEREO | **21.67%**           | **6.67%**         |
>
> These results reliably reinforce that our approach offers a simple and effective way to enhance unlearned models.
>
> ---
>
> > ## 3. Prior knowledge of the attacker’s vector
>
> SubDefense does **not** require any prior knowledge of the attacker’s vector or strategy.
> Its projection directions are obtained entirely from **SubAttack** during our own probing of the unlearned model.
>
> ---
>
> Thank you again for the helpful feedback. We hope the clarifications resolve the concerns raised.

---

### Official Review · Reviewer_Jfmd · 2025-10-29

**Soundness:** 3
**Presentation:** 2
**Contribution:** 2
**Rating:** 4
**Confidence:** 4

**Summary:**

This work presents a new white-box attack (*SubAttack*) against concept erasure methods and a complementary erasure method (*SubDefense*). The attack is designed to be more interpretable by learning adversarial inputs as optimized linear combinations of existing token embeddings. These embeddings are compared against the ones derived from other white-box attacks (UnlearnDiff, CCE) and evaluated as a defense when applied via a proposed subspace projection of the token embeddings. Despite its appealing dual structure, the paper, unfortunately, lacks focus and misses important baselines. For example, it completely leaves more advanced adversarially robust erasure methods out of the picture that rely on adversarial fine-tuning (like Receler, AdvUnlearn, or STEREO). Nevertheless, this paper explores a generally interesting perspective, but feels outdated with its restriction to linear combinations of existing tokens, while even more thorough erasure approaches like Receler or AdvUnlearn have already been proven to struggle with white-box jailbreaking attempts. Ultimately, the effectiveness of SubAttack and SubDefense are constrained by their linear nature, limiting the relevance of both to the community as already more potent defenses and attacks exist. If the experiments were extended to show that their SubDefense's projection approach (with UnlearnDiff, CCE, or SubAttack attack embeddings) is a successful orthogonal method that can be applied together with adversarial fine-tuning of the model, this paper would suddenly become a lot more relevant.

**Strengths:**

- (S1) **Well written** work with an interesting dual structure by proposing both a linear attack and a linear defense.
- (S2) **Appealing new perspective**: This work studies the composition of attack embeddings as linear combinations of other, non-erased concepts, and then uses safe subspace projections as a simple defensive mechanism.
- (S3) **Honest mention of the limits of the linear framework** in which SubDefense operates as a defense against non-linear attacks.
- (S4) **Extensive supplemental material** with details on more experimental results and ablations.

**Weaknesses:**

I find the following list of things to be major weaknesses:
- (W1) **No comparison to "non-linear" defenses beyond RECE** such as AdvUnlearn, Receler or STEREO. I understand that these models do not rely on subspace projections in the token embedding space like SubDefense (or the way they apply CCE or UnlearnDiff). However, adversarial gradient-based fine-tuning of model weights, such as STEREO or AdvUnlearn, is a relevant baseline from my perspective that the linear SubDefense framework should be compared to. The question here is whether the subspace projection (used throughout this paper, even with CCE) is a better or comparable defense than a costly inner adversarial loop (like STEREO, which uses CCE internally)?
- (W2) **Unsubstantiated argument**: The argument (in line 468) that defenses against adversarial attacks like CCE are largely unexplored is simply not True. For example, STEREO proved to be robust even against it.
- (W3) **UCE is generally not a very robust baseline** as it is very easy to circumvent. Showing the superiority of SubDefense against RECE is indeed interesting, but unfortunately, not a very relevant contribution in itself. Of course, this entire work "lives" within the linear framework, and RECE is (to the best of my knowledge) the only linear defense framework so far. However, the picture painted in this study should be a bit more complete by quantitatively demonstrating the effectiveness of SubAttack against STEREO.
- (W4) **The results for SubDefense on ESD (Table 5)** reveal that SubDefense fails to defend against CCE (not surprisingly), but crucially, even against its linear complement, SubAttack. Is SubAttack or SubDefense now not effective?

And these are minor weaknesses:
- (W5) **Unclear motivation for interpretability**: The claim that interpretability is important to control the robustness of unlearned deep generative models lacks clarity for me. It definitely is fun to look at word clouds, but the information in it is not necessarily very useful.
- (W6) **The results are spread across too many tables and figures**, making each part feel shallow and the overall picture hard to see. This paper needs more focus and clearer storytelling for the reader.

**Questions:**

- (Q1) A better wording in Line 204 might be: "non-negative representation" -> "non-negative linear combination"
- (Q2) Line 269: What is the intuition behind this additional defense step for CCE?

---

> ### Author Response · Authors · 2025-11-24
> **Author Response to Reviewer Feedback (1)**
>
> Thank you for the constructive and detailed feedback. We address each point below with new experiments and clarified explanations.
>
> ---
>
> > ## W1–W3. Clarifying Scope, Including Recent Unlearning Works, and Defense Setup
>
> ### **(1) Incorporating recent unlearning methods and new experimental results**
>
> We appreciate the reviewer highlighting adversarial-finetuning–based unlearning methods such as AdvUnlearn, Receler, and STEREO. These methods are indeed important, and we have now incorporated them into our analysis.
>
> Following the reviewer’s suggestion, we added **SubAttack (K = 5)** results on **AdvUnlearn** and **STEREO** for the erased concept *nudity*. The results are summarized below.
>
> | Unlearning Method | SubAttack ASR (K = 5) |
> |-------------------|------------------------|
> | AdvUnlearn        | **93.33%**             |
> | STEREO            | **21.67%**             |
>
> These results show:
>
> - **SubAttack on AdvUnlearn** → high ASR, indicating that even strong nonlinear unlearning pipelines still retain a *recoverable linear component* of the erased concept, despite UnlearnDiff being used inside the unlearning loop.
> - **SubAttack on STEREO** → lower but still non-negligible ASR, showing that a *small but interpretable linear subspace* persists even under robust adversarial unlearning with CCE in the loop.
>
> These findings reinforce the key message of our work: **linearly composed implicit concept directions may remain in a subspace after unlearning**, making interpretability-based analysis both informative and complementary to atatcks such as UnlearnDiff and CCE.
>
>
>
> ---
>
> ### **(2) Why UCE + RECE are the primary defense baselines**
>
> Our defense part, inspired by the interpretable diagnosis of unlearned models, focuses on simple **post-unlearning refinement** after the attack, not on designing a new adversarial unlearning algorithm (Section 2.2, *Defense*). SubDefense is intended to be a **lightweight, plug-and-play refinement** applied *after* any unlearned model.
>
> In this regime:
>
> - **UCE** remains a widely used base unlearning model.
> - **RECE** is a prior method designed to **further modify the already-unlearned model UCE** without re-tuning a base model.
>
> Thus, the UCE/RECE comparison follows the same *refinement setting* and is chosen due to shared design and scope, not due to linearity.
>
> In addition, we now provide **SubDefense results on STEREO**, showing that even a strong unlearner can be *further improved* using the residual subspace identified by SubAttack. For the concept *nudity* with 20 projected directions, SubDefense reduces ASR substantially:
>
> | Model  | ASR Before Defense | ASR After Defense |
> |--------|----------------------|-------------------|
> | STEREO | **21.67%**           | **6.67%**         |
>
> These results demonstrate that our approach offers a simple way to further enhance unlearned models by leveraging the interpretable diagnosis of implicit remaining concepts. We provide fuller discussion of the logic flow and motivations below.
>
>
> ---
>
> ### **(3) Clarifying the defense motivation: interpretability → diagnosis → refinement**
>
> The core contribution of our work is an **interpretability-driven workflow**:
>
> 1. **Diagnose** the failure: SubAttack uncovers the linear residual concept subspace via non-negative, sparse token mixtures.
> 2. **Reveal** the structure: these directions correspond to meaningful implicit cues (synonyms, context words, stylistic components) that remain after unlearning, composed in a linear way.
> 3. **Remove** the structure: SubDefense projects out exactly this subspace after diagnosis, yielding a simple, model-agnostic refinement.
>
> This pipeline illustrates that interpretability is not just descriptive—it directly enables the design of **simple, safe, plug-and-play corrections**. We hope this encourages future exploration of **combining linear and nonlinear interpretability structures** for even stronger refinements.

---

> ### Author Response · Authors · 2025-11-24
> **Author Response to Reviewer Feedback (2)**
>
> > ## W4. Why SubDefense does not block SubAttack completely
>
> SubDefense removes the **linear** residual subspace revealed by SubAttack. After applying SubDefense, we always re-run SubAttack on the defended model to identify *new* attack token combinations. The resulting ASR is consistently and substantially lower than before defense, demonstrating that removing the discovered subspace is effective.
>
> However, unlearned models still retain **nonlinear** traces of the erased concept. Because SubAttack operates under a linear formulation, it can only **approximate** these nonlinear remnants using linear combinations of tokens. This approximation can still produce a limited attack effect, which explains why the ASR does not drop to zero.
>
> This behavior is consistent with our stated limitation:
> **SubDefense eliminates linear residuals, but has difficulty fully removing nonlinear concept representations that survive unlearning.**
>
>
> ---
>
> > ## W5. Clarifying the role of interpretability
>
> Interpretability is central to our motivation. We first demonstrate that a **linear structure** can meaningfully recompose the residual concept, confirming that the erased concept persists in a structured and analyzable form. Within this structure, SubAttack’s non-negative, sparse token combinations reveal **which semantic fragments** remain (e.g., synonyms, stylistic cues, contextual activations), enabling both qualitative and quantitative diagnosis.
>
> This interpretable residual structure directly motivates SubDefense: the defense uses **orthogonal projection** to remove **the specific linear subspace** uncovered by SubAttack, guided by the observed linear concept-representation structure. Thus, interpretability is not merely descriptive—its structure directly enables evaluation and refinement.
>
>
> ---
>
> > ## W6. Improving clarity and focus
>
> We appreciate the reviewer’s suggestion. The paper covers three components—(1) linear-structure interpretability, (2) attack effectiveness and transferability, and (3) defense evaluation—which naturally require multiple figures and tables. In the revised version, we streamline the main text and move less essential visualizations and ablations to the appendix, making the core narrative clearer and more focused.
>
>
> > ## Other Questions
>
> **Q1.** Updated to *“non-negative linear combination.”*
>
> **Q2. What is the intuition behind this additional defense step for CCE?**
>
> CCE’s learned embedding is not restricted to the model’s vocabulary, so we directly project the CCE embedding to remove the target concept subspace identified by SubAttack.
>
> ---
>
> Thank you again for the valuable feedback and thoughtful suggestions.

---

### Meta-Review · Area_Chair_n8cR · 2026-01-07

**Summary:**

The remaining concern is the weak motivation on why the interpretation is needed in attacking unlearning models, which leads to additional concerns on insufficient comparison (e.g., missing strong defense baselines). Thus, eventually the reviewers did not confidently buy the paper.

**Reviewer Concerns:**

**Reviewer Jfmd**:
The reviewer shared five major concerns, which are partially addressed as follows:
1. (*weak baselines*) No comparison to "non-linear" defenses beyond RECE such as AdvUnlearn, Receler or STEREO – partially addressed by providing comparison to AdvUnlearn and STEREO (but no comparison to Receler) and showing that STEREO is better than the proposed defense under the proposed SubAttack (even in the case that STEREO is unaware of  SubAttack).

2. (*unsubstantiated argument) The argument (in line 468) that defenses against adversarial attacks like CCE are largely unexplored is simply not True – unaddressed

3. (*weak baseline*) UCE is generally not a very robust baseline as it is very easy to circumvent. – addressed by highlighting that the paper only considers the simple post-unlearning refinement methods, where UCE remains a widely used base unlearning model.

4. (*weak experiment results*) The results for SubDefense on ESD (Table 5) reveal that SubDefense fails to defend against CCE (not surprisingly), but crucially, even against its linear complement, SubAttack – partially addressed by claiming that it is an expected result but does not mention CCE results.

5. (*weak motivation*) Unclear motivation for interpretability for robustness – did not directly address this concern.

The outstanding and remaining concern includes the weak motivation – in attacking unlearned models, the interpretation of the attack does not really matter as long as the attack is successful; often the defense against interpretable attack is weak so the interpretability does not advance the frontier of the robustness of unlearning methods.

**Reviewer SBEJ**:
The reviewer raised three concerns, which are all addressed.
1. (*limited novelty*) SubAttack might be an iterative extension or multi-token variant of the previous approach rather than a fundamentally new contribution which raises some concerns regarding the novelty and the strength of the claimed contribution – the contributions are clarified.

2. (*limited comparison*) The paper compares SubDefense with only a few baseline defense methods and under a relatively small number of attack scenarios – addressed by highlighting that the paper only considers the simple post-unlearning refinement methods, so the baseline defense methods seem limited.

3. (*impractical assumption*) Does SubDefense require prior knowledge of the attack vector v_{att} used by the adversary? If so, this assumption might be unrealistic in practical settings – addressed by clarifying “”SubDefense does not require any prior knowledge of the attacker’s vector or strategy”.

No outstanding concerns.



**Reviewer 76Nh**:

1. (*weak novelty*) Regarding the novelty of the method, although the concept of inserting v_att tokens is employed, this attack and defense strategy appears to essentially be a linear orthogonal transformation – properly addressed by providing the value of linear orthogonal transformation of erased concepts (e.g., “Interpretable structure discovery: Section 4.2 shows that erased concepts persist along a coherent, human-interpretable linear subspace”)

2. (*instability of attacks*) The method still requires training an MLP, which is not training-free, casting doubt on the stability of the attack strategy – misunderstanding is clarified (i.e., “we do not claim SubAttack is training-free or unstable. Rather, our claim is that the required training is lightweight and stable in practice.”)

3. (*scalability issue*) The scalability of the method is questionable, as restoring a single concept requires extensive training – misunderstanding is clarified (e.g., “SubAttack requires far less training than adversarial attack baselines.”)

4. (*limited baseline*) The baselines compared in Figure 8,9 and Table 4 only include ECE and RECE, which seems to lack reliability –  addressed by highlighting that the paper only considers the simple post-unlearning refinement methods, so the baseline defense methods seem limited.

No outstanding concerns.

**Reviewer Scores:**

**Reviewer Jfmd**:
Final expected rating: 4 / final expected confidence: 4 – The rebuttal only partially addressed the raised concerns, so the reviewer would maintain scores.

**Reviewer SBEJ**:
Final expected rating: 6 / final expected confidence: 2 – The concerns are all addressed but due to the low confidence, I expect that the reviewer does not increase the rating more.

**Reviewer 76Nh**:.
Final expected rating: 4 / final expected confidence: 4 – The concerns are all addressed so the reviewer might increase the rating from 2 to 4.

---

### Decision · Program_Chairs · 2026-01-26

Reject